# CRISPR and Artificial Intelligence in Neuroregeneration: Closed-Loop Strategies for Precision Medicine, Spinal Cord Repair, and Adaptive Neuro-Oncology

**DOI:** 10.3390/ijms26199409

**Published:** 2025-09-26

**Authors:** Matei Șerban, Corneliu Toader, Răzvan-Adrian Covache-Busuioc

**Affiliations:** 1Puls Med Association, 051885 Bucharest, Romania; mateiserban@innbn.com (M.Ș.);; 2Department of Neurosurgery, “Carol Davila” University of Medicine and Pharmacy, 050474 Bucharest, Romania; 3Department of Vascular Neurosurgery, National Institute of Neurology and Neurovascular Diseases, 077160 Bucharest, Romania

**Keywords:** CRISPR genome editing, artificial intelligence, precision neuroregeneration, spinal cord injury, glioblastoma, digital twins, translational neuroscience, personalized medicine

## Abstract

Repairing the central nervous system (CNS) remains one of the most difficult obstacles to overcome in translational neurosciences. This is due to intrinsic growth inhibitors, extracellular matrix issues, the glial scar–form barrier, chronic neuroinflammation, and epigenetic silencing. The purpose of this review is to bring together findings from recent developments in genome editing and computational approaches, which center around the possible convergence of clustered regularly interspaced short palindromic repeats (CRISPR) platforms and artificial intelligence (AI), towards precision neuroregeneration. We wished to outline possible ways in which CRISPR-based systems, including but not limited to Cas9 and Cas12 nucleases, RNA-targeting Cas13, base and prime editors, and transcriptional regulators such as CRISPRa/i, can be applied to potentially reactivate axon-growth programs, alter inhibitory extracellular signaling, reprogram or lineage transform glia to functional neurons, and block oncogenic pathways in glioblastoma. In addition, we wanted to highlight how AI approaches, such as single-cell multi-omics, radiogenomic prediction, development of digital twins, and design of adaptive clinical trials, will increasingly be positioned to act as system-level architects that allow translation of complex datasets into predictive and actionable therapeutic approaches. We examine convergence consumers in spinal cord injury and adaptive neuro-oncology and discuss expanse consumers in ischemic stroke, Alzheimer’s disease, Parkinson’s disease, and rare neurogenetic syndromes. Finally, we discuss the ethical and regulatory landscape around beyond off-target editing and genomic stability of CRISPR, algorithmic bias, explainability, and equitable access to advanced neurotherapies. Our intent was not to provide a comprehensive inventory of possibilities but rather to provide a conceptual tool where CRISPR acts as a molecular manipulator and AI as a computational integrator, converging to create pathways towards precision neuroregeneration, personalized medicine, and adaptive neurotherapeutics that are ethically sound.

## 1. Introduction

Repair of the adult central nervous system (CNS) has generally been limited by the constraints, rather than the possibilities, of therapeutic intervention. Axons transected in the spinal cord retract with dystrophic end bulbs; astrocytes undergo a reactive response and form glial scars; oligodendrocyte debris becomes a binding partner of myelin-associated inhibitors such as Nogo-A, MAG, and OMgp; and the extracellular matrix provides a reservoir of chondroitin sulfate proteoglycans (CSPGs), which cement the non-permissive state [1,2]. More importantly, intrinsic transcriptional programs that could enable regenerative growth were actively suppressed by a myriad of regulators, such as PTEN, SOCS3, and KLF family members, while corresponding epigenetic landscapes associated with regrowth and plasticity were maintained in a repressive state [3]. Clinical experience has highlighted the reality of these roadblocks; neurological function lost after a traumatic spinal cord injury (SCI), progressive neurodegeneration, or invasive malignancy such as glioblastoma (GBM) is only very rarely restored with any contemporary treatments [4]. Neuroprotective pharmacological agents might delay an irreversible neurodegeneration but rarely restore lost function; trophic factors delivered through exogenous delivery vehicles only diffuse to a small volume in the target area and are lost through metabolism or clearance before substantive circuit reconstruction occurs; stem cell grafts are lost through lack of survival or meaningful integration or through immune rejection; and biomaterial scaffolds might support some structural regeneration but are incapable of re-establishing the precise long-range connections required for recovery of meaning. In these paradoxical circumstances, we define the historical stalemate confronting neuroregeneration [5,6].

Nevertheless, in the past two years there has been a rapid acceleration in two domains that now independently and quickly intersect directly with these deeply rooted issues: genome and transcriptome editing, using CRISPR systems, and computational inference, using artificial intelligence. CRISPR systems have matured into a toolbox in which their modalities can be used in concert [7]. Engineered high-fidelity Cas9 variants have been designed to maximize on-target activity while also minimizing off-target cleavage activity, thereby allowing for applications to sensitive neuronal genomes where any collateral damage might be disqualifying. Additionally, Cas12 systems, which have different PAM selections for greater targeting flexibility, and Cas13, which only works on RNA, would enable reversible, transient edits that do not alter the genome [8,9]. Base editors allow for precise repurposing of single nucleotides without double-strand breaks, and prime editors allow for templated changes to DNA without donor DNA and, thus, a reliance on donor DNA [10]. The Cas13 spatial RNA transport studies developed in 2024 and 2025 showed that transcripts could be actively relocalized to neuronal compartments within experimental systems using programmable messenger RNA “zip codes,” and had downstream functional implications for axonal regeneration in injury models [11,12]. This supports that there is some undeniable therapeutic leverage within the CNS beyond simply which genes are expressed; namely, related to the location of where the products of that expression accumulate in the cell, which is largely invisible with conventional strategies [13].

Additionally, artificial intelligence has evolutionarily pushed computation from retrospective classification to now prospectively orchestrating complex biological interventions. Deep learning models, trained on tens of thousands of perturbations, are increasingly predicting guide efficiency, isoform specificity, and collateral risk with an increased resolution of relevance, particularly obvious for RNA-targeting editors where both experimental and sequence contexts can have a strong bearing on efficacy [14]. For combinatorial CRISPR technologies that use active-learning frameworks, they suggest the most informative perturbations in an iterative manner, allowing rapid risk convergence onto synergistic sets of genes when searching within search spaces that are truly complex. At the scale of clinical decision-making, we now have rapid intraoperative imaging modalities such as stimulated Raman histology and, more recently, deep learning convolutional neural networks, which allow for margin-level diagnoses within minutes and allow surgical audiences to change resections in real-time [15]. Mathematical models of GBM growth, based on serial imaging, have been parameterized in individual patients, creating a primitive form of digital twins based on the future evolution of the tumor with different treatments. In SCI, similar models utilizing machine learning applied to clinical and imaging data acutely have also provided calibrated predictions of interventions associated with ambulatory outcomes 1 year later. Relevant predictors of recovery and trial stratification have also included features of interest [16,17]. Collectively these early efforts lay out the basic building blocks of a closed-loop therapeutic system composed of layers. At many levels, scaling from single-cell transcriptomes and spatial proteomic maps to intraoperative imaging and electrophysiological readouts, there is a sensory layer for state variables found. A model layer that translates data into predictive targets and control policies is needed using learned constituent relationships and mechanistic constraints [18]. An actuation layer performs the intervention based on CRISPR effector activity by modality, i.e., correction of transgenic DNA, transcriptional activation or repression of transgenes, RNA knockdown, or transcript relocalization, each of which has assigned vectors and routes of delivery to maximize distribution and safety [19]. A governance layer imposes constraints on safety, cites explanatory predictions, quantifies uncertainty, and provides human oversight. The synthesis of each of these layers is more than aspirational; elements exist now and in the primary literature from 2024 to 2025. Confirmations of RNA relocalization by adeno-associated virus (AAV) tagged to neurites with programmable editors represent spatial actuation [20]. The uncertainty associated with AI models for programmed prediction of guides can restrict design decisions to high-confidence regimes. To communicate information on comparative AAV variants used head-to-head in cis studies in the mammalian brain, quantitative comparisons can offer regional guidance on transduction efficiency [21]. Intraoperative deep learning workflows and patient-specific tumor models illustrate real-time state estimation and projection. Prognostic models in SCI mean we can clarify realistic endpoints and give meaning to trial-ready surrogates [22].

Implications are profound. First, spatial granularity has appeared in therapeutic design: interventions can modulate the subcellular localization of RNA, resembling the existence of zones of regeneration, rather than regarding neurons as identical transcriptional blocks [23]. Second, combinatorial coverage is coming into practice: the challenge of rationally testing the thousands of possible pairs of genes is minimized by AI-driven prioritization to limit laboratory efforts to narrow edges of the design space that are most likely to have a synergistic benefit [24]. Third, delivery is moving from generic to empirical: the comparative study of viral vectors in brain regions is producing quantitative maps to inform both the serotype and mode of injection rather than relying on a posteriori findings [25]. Fourth, time is an explicit variable: intraoperative AI is now signaling decisions with minutes over outcomes with prognostic models mapping potential impacts over months, meaning interventions can be incorporated into adaptive frameworks, which incorporate multiple-time horizons [26].

Framing each of the observations outlined above into meaningful scientific constructs can be achieved by linking evidence from original studies published in the years 2024–2025 that collectively demonstrate how recent CRISPR modalities and computational methods may be mapped in relation to the long-recognized biological limitations of CNS repair. The following sections will be arranged so as to provide a gradual mapping of this terrain. Section 2 examines intrinsic and extrinsic barriers to regeneration and emphasizes where barriers might overlap with measurable signals and putative molecular targets. Section 3 will map the latest innovations in the CRISPR toolkit, and Section 4 will address delivery systems while including spatiotemporal control. Section 5 and Section 6 will focus on the two areas that are being most actively explored—spinal cord regeneration and neuro-oncology. Section 7 will explore other neurological disorders, and Section 8 will propose a potential CRISPR–AI pipeline. Section 9 will discuss ethical, regulatory, and equity issues, and Section 10 aims to suggest notable directions for the coming decade. Overall, it is not our intention to provide prescriptive accounts but to trace recent developments and indicate how the field is developing the basis for data-informed, adaptive pathways for precision neuroregeneration.

## 2. The Barriers to CNS Repair: Molecular, Cellular, and Systems Landscapes

Regenerating the adult CNS will inevitably come up against a complex, layered inhibition landscape. While injured peripheral nervous system (PNS) axons are able to grow and eventually regrow and reconnect to original, functioning targets, CNS neurons tend to pause and ultimately enter a “conservative” state, conserving both energy and connectivity with metabolism as the priority and plasticity as a secondary [27]. Emerging work has established that limitations around regeneration are not simply singular but collective; layers of interconnected interactions around common barriers—transcriptional, metabolic, extracellular, immune, vascular, and systemic—are preventing CNS regenerative potential. These components are near seamlessly interrelated and thus the stabilization of a non-regenerative stage is inhibited [28]. Achieving the programmable capability to be regulated and to regulate through precision molecular tools or computational systems approaches leads us to a question: What does it look like to integrate?

### 2.1. Intrinsic Suppression of Neuronal Growth

One, adult CNS neurons appear, in comparison to their development, to be able to achieve far less intrinsic growth. To emphasize this point, recent single-nucleus transcriptomic attempts reported in 2024 showed that while axotomy-associated growth programs were up-regulated post-injury in corticospinal tract neurons, at analogous time points in peripheral axotomy injuries, there was a larger absolute up-regulation in gene expression [29]. Particularly, growth-associated regulatory programs such as Sox11 and KLF7 required for lengthening were down-regulated past functional length, and a variety of inhibitory signals from PTEN, SOCS3, and KLF4 mediated sustained inhibition of the regenerative pathways to mTOR and STAT3 [30]. The recent data also identified some additional layers of control. In a genome-wide screen of the axonal growth candidates, a vesicle-traffic protein, synaptotagmin-4, was the unanticipated candidate that defined the presynaptic machinery role regulating regenerative potential [31]. Mitochondrial dynamics build on this idea; injury-responsive mitochondrial dynamics in the axon growth cones were limited, and the chronic fragmentation of mitochondrial networks and new mitochondria due to defective mitophagy. The lack of ATP decreases availability as well and leads to an increase in oxidative stress. Compounding the problem, the neurons also have built up residual DNA damage, most likely tethering cellular programming toward viability maintenance rather than regeneration [32,33].

At the RNA level, long non-coding RNAs and circular RNAs reinforce inhibitory states. A lncRNA–miRNA–kinase feedback loop from 2024 understanding elevated apoptosis and inflammation signaling occupied across the levels of injured tissue while highlighting a mechanism in which post-transcriptional circuits (e.g., microRNAs) can enforce and overcome maladaptive responses [34,35]. In total, this work specifies the components of an “intrinsic suppression program” in which transcriptional silencing, metabolic derailment, genomic instability, and regulation of the non-coding RNAs converge into interdependencies that constrain competent growth [36].

### 2.2. Extracellular and Immune Barriers to Repair

The microenvironment surrounding the tissue biology post-injury serves as a biochemical and biomechanical barrier, which promotes a neuronal quiescent state. Reactive astrocytes proliferate to create a glial scar and secrete CSPGs, e.g., aggrecan, neurocan, and versican. CSPGs bind the PTPσ and LAR receptors of the neurons and activate downstream RhoA–ROCK signaling, which collapses growth cones [3]. Spatial transcriptomics from 2024 shows, regardless of learning and behavior, that reactive astrocytes are heterogeneous populations, with reactive astrocytes displaying features of protective programs vs. reactive astrocytes that were labeled with inhibitory matrix elements. Also, it is worth considering the location where reactive astrocytes originate to ensure that more parenchymal-derived astrocytes have greater pro-regenerative responses than glial-derived astrocytes, but both sets of astrocytes still have the same generic injury cues [37]. Scar structures are active systems that must be maintained. The activity of hnRNP U that regulates astrocyte proliferation and the establishment of scar boundaries showcases a repertoire of astrocyte function—growth cone collapse (scar) vs. growth cone capture (medium to result) [38]. Rab27a-mediated vesicle release transfers CSPGs for release into the extracellular space and indicates that transfer to biologically active tissues is not passive release of the inhibitory molecules but a continuous release of inhibitory molecules into the exterior [39]. The mechanical properties of the scar are important, too! In 2025, rigid biomaterials that tried to emulate the native tissue organization and various stiffnesses (which decreased gliosis and endothelialization in quantified analyses) discernibly indicated potential for the neurons to detect the mechanical as well as chemical characteristics of the remediated space [40]. Myelin debris presents yet another chronic impediment. There is an array of ligands (Nogo-A, MAG, OMgp) from the boundaries of the lesions that move with their representative NgR1–LINGO-1 complexes to start collapsing signaling. Microglia have opportunistic roles as debris phagocytes and defend-contacting axons during the acute time point after injury, which was shown to be an advantageous relationship during acute time frames when they are successful at phagocytosis because of their cytoskeletal remodeling trajectories becoming neutralized and less stable [41]. Studies have shown the component of phagocytosis via fascin-1, the phosphorylated form of fascin-1 a member of the fascin family of actin-busting proteins, providing a rationale for the role of actin dynamics in the debris phagocytosis [42]. A 2024–2025 imaging study with a positron emission tomography (PET) tracer designed to identify where myelin is actually residual in units of rodents and humans has provided a candidate for a translational biomarker to substantiate the inhibitory burden [43,44,45].

The activity of immune cells supporting the tissue can further modulate the environment. Microglia changes can be detected depending on the time phase and location: phase-acute containment of damage, behavior-beneficial; phase-chronic, behavior-microglia become pro-inflammatory cells and begin stripping out synapses. In experimental studies of various animal models, the use of a chemokine gradient to prolong linger time at the core of rodent lesions mitigated neuronal cell death and enhanced repair of the neuronal population [46]. Macrophages provide oxidative stress and contributions, including ferroptosis. Some studies within the last two years have shown improvements to functional recovery associated with inhibiting the ferroptotic pathways in rodents [47]. There are some remarkable synergies with GBM. It binds matrix CSPG and collagens and is accompanied by increases in tissue stiffness; it forms neuron–tumor synapses via AMPA receptors–tumor siphoning activity whilst silencing parental neurons. These tumor-associated adaptations emulate states of chronic injury and imply the non-resiliency to repair from cancer and trauma is indicative of converging extracellular modalities [48].

### 2.3. Systems–Level Stabilization of Non-Regenerative States

Epigenetic influences and systemic characteristics could be extenuating non-resilient modes beyond the local injury. Overall, based on chromatin work, there appears to be a robust closing of loci associated with growth via educational enrichment of repressive histone proteins mediating transcriptional regulation, such as H3K27me3, as well as long non-coding RNA molecules like MALAT1 and NEAT1, influencing both pro-inflammatory and apoptosis progressions, while alternative splicing creates cytoskeletal isoforms to the inhibitory state. Together, these represent contributions to something referred to as a “molecular record of injury,” whereby maladaptive cell states are sustained in the chronically traumatized tissue in a transcriptional and epigenetic sustained manner [49].

Observe vascular-related dysfunction contributes to sustained stable states through regions out of the acute phase of injury. For example, it is generally accepted that the blood-spinal cord barrier is not only compromised in the acute injury phase but remains compromised in “breached” conditions exposed to the infiltrating parts of fibrinogen, revealing pro-inflammatory properties from astrocytes, and may even activate microglial properties [50]. In 2025, a group demonstrated endothelial integrity could be reverted in a chronic injury using MerTK signaling to recover vascular flow integrity reduce permeability and levels of inflammation, while outcomes were also improved therapeutically through administering imatinib, a PDGFR-mediated chromatin remodeling pharmacologic silencer [51]. Also worthy of note, assessments of chronic vascular flow impairments using MRI revealed values for long-term recovery that would associate with peripheral flow physiology [52]. Moreover, systemic signals work to expand these stable barriers to all domains of organisms. Age and sex of the human examined influence the glial resilience and myelin density per area, which shape the pre-existing level of regenerative potential in primate and human spinal cords [53]. Exosomes from circulating immune cells carry inhibitory miRNAs into the CNS. Gut/community dysbiosis alters organismal cytokine state and inflammatory tone. Collectively, these works move the injured CNS from localized injury to a more expansive physiological network within which peripheral systems will ultimately govern the central repair state [54]. What emerges, when considered together, is a pathologically self-enforcing inhibitory ecosystem. Chronic residual debris overlaps progress and exposes cumulative inflammatory states, leaking compromised barrier zones to form scarring, and the circuitry of non-coding RNA reinforces transcriptional repressions. The complexity is again linked to the concept that one intervention, such as the deletion of PTEN or digesting CSPG, affords only incremental positive forward progress toward recovery. It is clear that sustainable repair needs to be multiplexed systems-level repair engaging all levels of the repair spectrum: molecular, cellular, and systems [55].

Recent work has led to the conclusion that regeneration failure is a matter of interrelated barriers rather than isolated challenges. Neurons transcribe as cautious, extracellular, and immune components create potential cellular inhibitory niches, and systematic and/or epigenetic mechanisms stabilize these states into chronicity over time [56]. The notion of interrelated systems barriers offers a much-needed conceptual platform for the next aspects of this review—productive manifestations of CRISPR systems as programmable molecular tools and artificial intelligence as compound, multi-level signal integrators—to form an intentional adaptive repertoire of interventions to redefine the inhibitory nature of the CNS after an insult or injury [7]. To orient towards the unique complexities of repair in the CNS sphere, it will be useful to summarize the unique major molecular and cellular barriers blocking regeneration. Table 1 is intended to outline inhibitory pathways, cellular responses, and regulatory mechanisms recently mapped from the published literature and new strategies that may be used to manipulate them. It does not represent a completely exhaustive list but rather a synthesis of illustrative examples retrieved from 2024 to 2025 characterizing both the array of barriers to regeneration and potential avenues for intervention.

## 3. CRISPR as a Molecular Toolbox for Neural Reprogramming

Components of the adult CNS can endure injury; however, repair entails treatments that can go beyond symptomatic modulation and actually change the molecular logic of neural cells instead. On account of their programmability and exponentially growing repertoire, CRISPR technologies now exist to do just that. Initially developed as nucleases to induce double-strand breaks, CRISPR systems have evolved into platforms that can cut, replace, regulate, and even rewrite the epigenetic landscape [70]. In the context of CNS injury and tumor sustainability, CRISPRs represent tools ranging from removing the inhibitory bottlenecks to creating newfound regenerative states entirely from scratch [71]. The following sections will attempt to align recent innovations related to editing types, functional discovery, delivery advances, and translational use and to elucidate how CRISPR can revise both neurons and their microenvironment.

### 3.1. Expanding the CRISPR Toolbox for Neural Systems

More compact and precise CRISPR nucleases have been an important innovation for neural applications given the constraints of delivery capability and cell sensitivity. The ability to package editing complexes into single AAV vectors, enabling delivery of compact Cas12a variants and ultra-miniaturized Cas12f nucleases, allows us to circumvent the capacity constraints we previously experienced, where we needed to use multiple vectors that caused us to reduce efficiency [72]. Similarly, are IscB-derived base editors that facilitate adenine and cytosine editing into ultra-compact scaffolds, leading to efficient correction in vivo using a single AAV system. The results greatly reduce the logistical burden of CNS delivery while taking advantage of compact enzymes that maintain the precision necessary to use safely in post-mitotic neurons [73]. Even beyond DNA, the editing landscape continues to expand into RNA. Cas13 systems have been developed for splicing repair, regulating translation, and programmed RNA modification. The Cas13 fusion constructs utilizing a chemical mark such as N4-acetylcytidine provide reversible control over transcript stability and localization. This capability is especially contextualized for neurons that engage structural plasticity with localized protein synthesis in axons and dendrites to react to injury [74]. In the event of injury, RNA editing provides a mechanism to preserve functionally viable areas (e.g., eloquent cortical regions) from permanent modulation and allows for reversible, conceptual changes during mechanisms of injury [75]. Approximately, in vivo studies using prime editing as a newly proliferating technology provided unprecedented levels of precision at the level of the DNA—one use provided evidence of repair for pathogenic Atp1a3 variants discovered using mouse models of alternating hemiplegia (an injury that can be debilitating), where molecular repairs were comorbid with functional restoration/behavior and increased models survival potential of an injury. For neurons, which are poor at homologous recombination, both advancements in prime and base editing present opportunities for long-lasting correction with fewer off-target effects [76].

The second class of CRISPR editing is epigenetic CRISPR editing. Using catalytically inactive Cas9 fused to either a transcriptional or chromatin-modifying enzyme, it is possible to non-reductively regulate growth programs. DCas9-p300, for example, has been used to reopen growth loci that are inhibited by H3K27 trimethylation, just as dCas9-TET1 has demethylated an axon growth gene promoter [77]. Epigenetic programming is not only non-reductive, but also a reversible and flexible development modality, which, in light of critiquing the neuroregeneration space, is translatable based on the fast-paced and sometimes unpredictable requirements of neuroregenerative environments as opposed to permanent edits for which risks are unknown [78]. Tying CRISPR strategies directly to the outlined constraints to regeneration discussed in Section 2, one can now reason a pathway map of molecular entry points. At the intrinsic level, knockout of PTEN removes inhibitory control on mTOR, and in the case of deleting SOCS3, the activation of additional STAT3-regulated transcription can be maintained. Editing the gene KLF4, disrupts the inhibition of growth via KLF4 where KLF7 can be activated in a CRISPR-dependent manner, endogenously promoting elongation competence. Mitochondrial modulation has emerged as a new axis: CRISPR editing of DRP1, MFN2, and OPA1 restored mitochondrial trafficking and function to locally affected ATP failure regenerative axons.

In this regard, we ought to target postsynaptic and presynaptic components of the extracellular interface (PTPσ and LAR chondroitin sulfate proteoglycan receptors, for example), which would ameliorate the inhibitory components of glial scars to treatment. Excitingly, recent reports of CRISPR loss-of function studies documented that deleting (in addition to these proteins) PTPσ, LAR, and others suffices to restore growth cone migration along CSPG-rich matrices [79]. The same scenario could be constructed for the potential limitations of ferroptosis regulators (e.g., GPX4 and FTH1) that may be regulating gene networks activated by macrophages: CRISPR deletion would disrupt lipid peroxidation driven by macrophage infiltration [80,81]. Similarly to this, CRISPR systems using RNA targeting have a similar story, as they can silence maladaptive long non-coding RNAs or microRNAs that stabilize inhibitory circuits (there are some long non-coding RNAs that amplify apoptotic and inflammatory signaling). These are direct modulations in post-transcriptional repression; meanwhile, regeneration must occur in the context of the CRISPR DNA editing interventions altering all dirt [82]. The tumor microenvironment is also a similar domain or target. GBMs employ similar tricks of injury scars: a dampening matrix to keep cells out of their environment, like secreting CSPGs, matrix stiffening, and synaptic takeover. CRISPR directed disarmament of its oncogenic drivers (e.g., EGFRvIII and IDH1-R132H), then a correction of donor genes related to DNA repair (e.g., MGMT), and then disabling of immunosuppressive checkpoints (e.g., PD-L1) all exhibited the ability to disrupt these established tumor resistance programs [83]. Patient-derived GBM stem cell-based CRISPR screens have shown the ability to target both shared dependencies and dependencies that were specific to each subtype of GBM cell type and then multiplexed these interventions to dynamically change their targets in real time to develop into a treatment option [84].

### 3.2. Functional Discovery Through High-Throughput Screens

The underlying strength of CRISPR is not so much autonomous editing as it is a discovery machine. Single-cell Perturb-seq is now being developed for in vivo use, so there is potential for pooled CRISPR perturbations to be read out in intact neural circuitries [85]. The application of this technique to brain tissue allowed us to uncover regulators of synapse maturation and neuroprotection that would not have been uncovered in dissociated culture systems. In assembloid systems of neurons, astrocytes, and microglia, CRISPR interference screens the glia-dependent modifiers that emerged as the most predominant modulators of neuronal resilience and resilience overall, highlighting the importance of profiling not just neuronal populations but also their supporters [86].

The Multiome Perturb-seq, which provides transcriptomic and chromatin accessibility readouts, delivered additional resolutions to help us determine which edits tend to have persistent epigenetic change versus transient transcriptional activity, which may not endure/bound during recovery paradigms. This is relevant for regeneration, as transient variations in gene expression may not be able to outcompete stable and/or entrenched states [87]. These functional-genomic formats are being adapted for use in patient-derived GBM organoids, where CRISPR libraries will incorporate the temporal resolution of the system to allow for tracking resistance dependency mechanisms in real-time. Collectively, these datasets create a discovery-to-intervention loop where mutations that emerge as significant modulatory genome editors can be pursued in vivo using the small editors and delivery modality now available [88].

Another important complementary piece is the use of mathematical models studying genome-wide perturbation datasets. Transcriptional control in neural cells is fundamentally stochastic and nonlinear, driven by bursty expression, chromatin gating, and feedback loops that determine if an edit was modified transiently or permanently. New modeling methods have begun to abstract and formalize these dynamics in order to capture genome-wide regulatory architectures while also predicting state transitions that have an order of granularity that conventional assay systems would omit [89,90]. The power of CRISPR pertur-seq combined with multiomic screens and math models enables the potential discovery of transcriptional attractor states that would be maximally permissive to regeneration while also predicting when those edits would likely revert to negative or inhibition pathways [91]. The combination of mathematical modeling plus CRISPR discovery and analytics combined with AI provides a potential pathway forward beyond cataloging differential expression to sculpting transcriptional states to induce sustained plasticity and functional repair [92].

### 3.3. Delivery and Safety Innovations

The translation/capitalization of CRISPR to CNS relies on delivery and safety. Enormous progress in AAV engineering has yielded capsids or vectors that are non-retroviral, have transferrin receptor targeting capability to cross the BBB (blood–brain barrier), and allow for effective, human and brain-wide expression efficiencies. These advances have been made with brain-targeting lipid nanoparticles, and systemic delivery of Cas mRNA/guide RNA to neurons and endothelial cells has the advantages of short-lived expression and repeat dosing [93]. In this context, there are now specialized AAV toolkits with tropism for glial subtypes and spinal interneurons for multiple SCI interventions, which expands the capability to clarify circuitry (and not only projection neurons) and support cells. Dual-action immune evasion strategies increase the feasibility of these interventions to be performed: deimmunized Cas proteins and stealth AAV variants could prevent mitigating protective host responses, while self-deleting editor systems could be used to limit prolonged exposure to nucleolytic action from host cells [94]. Biodegradable nanoparticles would also enhance the safety of chronic exposure, with existing assays indicating the probability of chronic inflammation is negligible. These safety innovations should not solely be considered but have utility as context enhancers because the requirements for enabling long-term competency in healthy neuronal functions, require interventions that correct molecular pathologies while not creating new potential risks [95].

### 3.4. Toward Integration with Artificial Intelligence

Social observers have previously indicated that CRISPR enables the reprogramming of molecular state, but the data that builds from high-throughput perturbation studies (albeit potentially terabytes of used memory) may well exceed the indigenous capability to conduct signaling analysis. New analytics and predictive AI are now commonplace, leveraging a growing public interest in and single-cell CRISPR screens to predict off-target effects from host genomes that become increasingly complex or managing library guide RNA from a more extensive list of potential glial subtypes, just to mention two of the possibilities [96]. In addition, AI may offer collaborative prospects for predictive modeling along similar lines and perhaps offer interventions initially precise at molecular space but eventually coherent at systems space in terms of designed interventions viewed from circuits converging multiplex edits [97].

These synergies would imply a closed-loop conception of CRISPR viewed as programmable interventions, while AI’s physical energies drive the analyses of cellular and systems responses regarding calibrating future models of gene and epigenetic editing. This convergence provides a useful conceptual explanation prior to addressing the focus of the next section, with AI as the integrator of the complexities of neuroregeneration [98].

CRISPR technologies have moved beyond blunt genetic scissors to programmable platforms that interrogate DNA, RNA, and chromatin at scales approximating the natural limitations of neural tissue. Their deployment in intrinsic inhibition and glial and naïve reprogramming, and barriers in the extracellular matrix, mitochondrial dysfunctions, and tumor microenvironments have potential examples of depth and scope [99]. The adoption of high-throughput CRISPR screening technologies has yielded systematic maps of regenerative competency that identify any transcriptional regulators, signaling pathways, and epigenetic programs that are architects of repair competency in a variety of neural systems. These resources provide novel opportunities for the community to more rapidly translate and safely apply advancements in biotechnology into clinical application. With artificial intelligence, it is possible that these artificial devices will go beyond merely blocking inhibition to actually building adaptation states of neuroregeneration [7]. Although the previous section has identified the principles and applications of CRISPR systems that operate in neural domains, it may be advantageous to provide extensive consideration as a conceptual context for the main platforms presently being explored using Table 2 to show a selection of representative modalities, the mechanistic basis, and one or two example studies for the recent years indicating how distinct strategies for editing could be applied to neuroregeneration and neuro-oncology contexts.

Figure 1 aims to provide a visual framework regarding the main CRISPR platforms and selected applications in neuroregeneration. The figure is intended to be a visual guide to the above, illustrating how various editing paradigms have been investigated for reactivation of intrinsic growth programs, inhibition of discouraging environments, reprogramming of glial fates, and inhibition of oncogenic pathways. The figure also includes discovery pipelines to highlight new targets but certainly does not intend to be exhaustive.

## 4. Artificial Intelligence as the Integrator of Neuroregeneration

The concept of repair in the CNS is a moving target; relating specifically, molecular dynamics can occur in minutes, tissue remodeling in days, renewed circuit activity in months, and clinical recovery or decline can span years. Today, the only practical means of addressing these inequivalent time scales with interconnected therapeutic strategies is artificial intelligence (AI). Rather than competing with molecular modalities, such as CRISPR, AI provides the framework from which molecular tools can be targeted, sequenced, and updated as the biology changes. The following synthesis is intended to portray many of the emergent aspects into a singular understanding of the role AI serves as the systemic interpreter and navigator of neuroregeneration and adaptive neuro-oncology.

### 4.1. Data Integration, Temporal Forecasting, and Network Reconstruction

The modeling architectures of contemporary machine learning, like transformers and graph neural networks, have incorporated multimodal data—genomics, single-cell transcriptomes, chromatin landscapes, proteomics, metabolomics, imaging, and electrophysiology—by embedding these forms of data into one shared latent space, which quantifies their interdependencies across multiple scales [107,108]. Models like this elucidate chromatin closures marking metabolic bottlenecks, cytokine gradients issuing glial states, and protein complexes dictating regenerative modules. Spatially regularized inference adds realism to the model by incorporating the micro-domains of CNS tissue, and perturbation-aware modeling relates expected outcomes exactly at CRISPR perturbation in a straightforward manner. Your target selection could, in part, functionally identify nodes with systemic relevance instead of just differential expression [109]. The temporal models extend the previous frameworks with respect to the nested phasic orderings that exist in any regenerative biology—epigenetic repression precedes metabolic activation, cytoskeletal assembly occurs before synaptogenesis, and microglial protection occurs before the debris-clearing effect begins to abate. Neural ODEs and Bayesian recurrent networks provide explicit scheduling of edits in sequence rather than indiscriminately stacked, minimizing intoxication and avoiding antagonism. In cancer care, clonal dynamics that are inferred from serial liquid biopsies and consider imaging may predict the trajectories of resistance clones, allowing proactive molecular maneuvering before clinical “escape” [110].

At the circuitry level, connectome-informed models may evaluate how axonal extensions will integrate and discern certain motifs that do or do not have the potential for maladaptive synchrony or stabilizing function. By including the dynamical systems of electrophysiological dynamics, predictions are guaranteed to be compatible with neurophysiology and not solely statistical association [111]. This allows regenerative programs to be congruent with realistic expectations of respectable network behavior, and with GBM models, it shows that connectivity is lost beyond the lesion core, thus redefining success as including not only cytoreduction but also preservation of cognitive networks.

### 4.2. Immune Microenvironments, Vascular Dynamics, and Glymphatic Function

The neuroimmune interface presents opportunity as well as barriers. Spatial AI can now resolve microglial and macrophage subtypes by transcriptome, but also their tissue geometry and neighbors, to predict when the inflammatory tone shifts from reparative to inhibitory, and in which niches [112]. These maps reveal actionable interactions—chemokines that trap microglia away from lesion cores, lipid-supply failures that allow ferroptosis to be sustained, and macrophage receptors that differentiate efferocytosis competency [113]. Predictive analyses like these allow CRISPR-type strategies to be spatially refined, enabling tuning of particular microenvironments while not repressing essential responses in general [114].

Virtually identical advancements in neurovascular and glymphatic informatics allow for parallel models. Deep learning and dynamic approaches can now track leakages across the blood–brain barrier via contrast-enhanced MRIs and ultra-fast ultrasound and quantify perfusion heterogeneity; we can also identify clearance via activated aquaporin-4 and/or glymphatic-mediated processes during dynamic–contrast imaging and water-clearance via > AD with diffusion–weighted sequences [115]. These factors are indicators for which regenerative interventions may fail due to edema, clearance failure, or vascular collapse. Mechanistically, these factors are associated with CRISPR targets, such as their effect on tight-junction regulators, aquaporin-4 polarity, endothelial stabilizers, etc. [116]. This situates the molecular editing in the context of the physiological realities of fluid and barrier and creates a multi-compartmentalization map, where immune, vascular, and clearance dynamics are not simply reflections but, not to overhype too much, determinants of regenerative viability [117].

### 4.3. Real-Time Biosignals, Brain–Computer Interfaces, and Behavioral Readouts

More than just biochemistry, AI has supported novel possibilities for reading direct functional recovery. Real-time streamed pipelines can classify intracortical and spinal signals live. The classifier assesses regenerative (unfolded directed information flow; adaptive cross-frequency coupling) and pathologic (pipe-like signatures; brittle high gamma bursts; runaway beta loops) patterns [118]. Closed-loop controllers can apply the classifiers to engage or modulate stimulation protocols or rehabilitation paradigms or terminate molecular interventions if a risk state has occurred [119]. Wearable or implantable sensors provide complementary pieces to track gait cycles, compensatory synergies, autonomic shifts, or usable patterns and thresholds for fatigue. Given these continuous streams, AI can be used to develop interpretive trajectories of functional loss or gains [120]. Adaptive BCIs can support recovery by shaping neural manifolds with respect to motor intent while establishing new circuits. Generative controllers may freely and safely explore stimulation parameter space that tests around parameter values shown to be effective while instantly rejecting unsafe regimes [121]. This brings us back full circle and upgrades the thinking to consider potential impacts of molecular therapies, which exist not just in histology or imaging but in lived minute-to-minute behavior [122]. Arguably the most impactful change is the introduction of personalized digital twins—computational avatars that combine multiple omic profiles, imaging, electrophysiology, and rehabilitation trajectories into a constantly adaptive model of an individual’s biology [123]. These twins combine mechanistic modules (e.g., axonal conduction, edema kinetics, and vascular flow) and learned components (e.g., cell-state transitions and tumor clonal dynamics) and can update hidden parameters as new patient data arrives. This twin can propose CRISPR intervention schedules for inducing regeneration, optimizing both efficacy and safety, while reporting confidence intervals, risk signals (e.g., stoplight-style), and “what-if” scenarios for alternative strategies [124].

Organoid platforms reinforce this by providing high-throughput outputs of CRISPR perturbations. AI closes the scale gap by mapping cell-state shifts and network motifs derived from organoids to predict in vivo circuit outcomes [125]. Developmental diffusion models trained on longitudinal single-cell data form in silico neurogenesis trajectories, enabling candidate editing programs to be assessed against idealized developmental arcs. Iterative loops of simulation, editing, measuring, and retraining allow multiplex interventions to be isolated that have the highest probability of integration-competent regeneration [126]. Large multimodal foundation models take hypothesis generation even further by embedding text, omics, imaging, and biosignals into concordant spaces. A self-supervised objective can identify rare regenerative phenotypes that are invisible to curated datasets, and causal discovery methods can capitalize on the perturbation data to distinguish drivers from passengers. Counterfactual simulations enable counterfactual simulations to evaluate how editing order will affect the individual’s outcomes and develop clinical plans based upon a mechanistic case of therapeutics, not, for example, association [127].

### 4.4. Clinical Translation, Trial Design, and Ethical Alignment

While attempting to translate composite neural regeneration projects into patients, AI is of some utility in trial design by grouping subjects in biologically consistent groups, using some adaptive design—varying dosing or inclusion criteria once early surrogates had been amended—and generating synthetic control arms in federated registries when randomization is not plausible [128]. The federated learning systems provide ways to coordinate across-center sites or institutions without needing to provide shared raw data, earning the privilege to coordinate across centers via domain adaptation and transfer learning, developing generalizable models [129]. We are not (yet) back to the early days of clinical and translational science when we had to manage even small events manually. Foundation models are computationally taxing; however, parameter-efficient tuning means foundation models can be tuned well, do not take much in load, and, more recently, distilled models perhaps offer a route for going bedside with the utilization of their implications.

Modern explainability offers not just a ranked list of features but a list of concepts that will have biological meaning for clinicians—scar-associated microglia, perivascular leakage, or resistant tumor clones. Counterfactual outputs allow, by identifying the least number of changes that would have resulted in a different recommendation, to provide transparency. Governance structures ensure auditing, trails, and versioning, and lock models while separating adaptive from locked models and prevent an algorithm build to think interpolatively about clinical practice without acknowledging it will be a concern. To potentially risk-proof the preface to reporting whether there is maladaptive plasticity, epileptogenesis, hemorrhage, or tumor escape, then possibly shift the decision to the plan that offers benefit and reduces worst-case harm [130].

### 4.5. The Implementation of Closed-Loop CRISPR–AI Systems

The trend of these product areas is moving towards closed-loop therapeutic systems. Sense integrates multimodal information through omics, imaging, connectomics, biosignal, and rehabilitation information into time-aligned stores. Infer applies combinations of temporal, spatial, causal, vascular, and immune modeling to identify targetable levers, contributing the best time to the models to prepare [131]. Act applies chemical and bioelectricity to return outcomes to the loop. Adaptations from current small-animal, organoid simulation will align with reduced discovery timelines; AI offers multiplex edits. CRISPR enacts multiplex edits, and predesignated modifications generate updates based on the transcriptomic and electrophysiologic changes. Our critical next step in simulating B2VOC is about moving models into patients, which is dependent on the in-setting of delivery being safe and the safety being scaffolded and embedded in the framework of a regulatory environment, but therapeutic integration principles should not be difficult to embrace: (a) be conservative with acting, when uncertainty is high, (b) diverge only when there is converging evidence, and (c) remain adaptive as the biology changes to the moment where preservation life intervention models become adaptive and permanent. In coherence, AI value can add back to neuroregeneration, circulating consciousness that can aggregate multi-scale in the client across every piece of explicable data into either a spatially and sort temporally ordered risk-aware plan based upon variability profiling; foresee where timing or what molecular lever will be most relevant to the system; observe/notice/detect when elements of the biological system are changing; and adapt based upon one or more of these obvious signals in action, based in consistency. Within this, CRISPRs would become less clinical tools and meaningfully more part of an adaptive therapeutic system—observing, inferring, acting, and learning within the dynamic disequilibrium of CNS repair as it is lived within. Table 3 tries to present exemplars of the use of AI in 2024–2025 across computational domains, methodological processes, exemplars of biomarkers, and impacts across domains of translation that were documented in the established definitional pulse of SCI and neuro-oncology. The aim is the expose methods to identify increases in propensity in their respective fields as a mechanism for universal convergence across as a cross-sectional method of understanding and not an exhaustive catalog.

Figure 2 aims to summarize the domains where artificial intelligence has been applied in neuroregeneration, showing how multimodal data can be integrated into predictive and therapeutic frameworks.

## 5. Convergence in Spinal Cord Regeneration

SCI clearly illustrates the total resistivity of repair in the adult CNS. The lesion site is a dynamic microenvironment with axonal inhibitory molecules, glial reactivity, vascular impairment, neuroinflammation, and metabolic collapse all concurrently involved and negotiating to prevent endogenous repair options. When we have endeavored to target them independently (e.g., neurotrophins, scaffolds, electrical stimulation, etc.), some progress has been made, but functional repair has not really developed [141]. With the introduction of CRISPR molecular editing, we are experiencing real-time SCIs updated to AI-informed integration, transmuting SCI from a static medical condition to a dynamic medical system that needs to be forecasted and monitored [142]. In the context of the AI long-term prognosis, regenerative strategies are adaptive rather than predetermined strategies, whereby we can combine orthogonal and advanced molecular, cellular, structural, and rehabilitation aspects into integrated adaptive therapeutic ecosystems and pharmacological modalities [143].

### 5.1. Models as Layered Testbeds of Convergence

Standardized rodent models for contusion and hemisection are still valuable for reproducible testing of CRISPR perturbations, but the nuances of human lesions can greatly benefit from both porcine and primate models, which can approximate spinal biomechanics and ratio scales of white and gray tissues and vascular vulnerability [144]. In these larger models, advances in imaging analyses with AI detection are applicable to observable (non-observable) radiomic features, such as microvascular rarefaction mechanisms of heterogeneous diffusion anisotropy—where both of these features correspond to regenerative capacity [145]. Non-mammalian models, particularly zebrafish and axolotls as regenerating benchmarks, have and continue to maintain integrity. Zebrafish and axolotl show spontaneous axonal regrowth and scar-free repair. Single-cell atlases from these models are being aligned with mammalian data, including AI-generated predictive structures, which have identified evolutionarily conserved pro-regenerative modules such as metabolic flexibility and cytokine switching with epigenetic openness—all of which are very useful for human editing [146]. Human spinal organoids and assembloids will help close the translational gap, enabling pooled CRISPR screens performed at scale and with single-cell resolution readouts. Their pairing with AI platforms will enable cross-species comparisons that go beyond descriptive biology to predictive road mapping for interventions across a widely divergent continuum of regenerative potential [147]. As an illustration of examples, CRISPR-directed molecular and metabolic reprogramming showcases a common theme in regenerative engineering: that they consisted of manageable constraints around oxidative and glycolytic metabolism impairments. Emerging in injured neurons, the impairment in metabolic energy depletion is an equally limiting factor as is signaling inhibition [148]. For example, CRISPR-based interventions will be evaluated targeting mitochondrial transcriptional regulators (e.g., PGC1α, TFAM) and a component of the oxidative phosphorylation electron transport chain to enhance oxidative capabilities, while targeting and editing the glycolytic enzymes (e.g., LDHA, PKM2) to achieve a balanced approach to energy flexibility with axonal energy demands. Together these aim to create neuronal metabolic environments that may themselves be regenerative bottlenecks amenable to editing within [149].

Vascular integrity is a key variable. Blood-spinal cord barrier (BSCB) opaqueness to injury remains therapeutically transient long after injury, which allows inflammatory infiltration and edema. Enhancement to barrier integrity is being evaluated for CRISPR-mediated edits to junctional endothelial proteins (e.g., claudin-5, VE-cadherin) and pericyte regulators [150]. Simultaneously, the application of AI to explore dynamic contrast MRI and ultrafast images to assess treatment risk factors, i.e., which patients are most at risk for vascular collapse. Both sets of work link CRISPR-related metabolic and vascular edits with clinically relevant imaging markers in research that shifts the focus of neuronal axonal repair to incorporate the systemic physiologic context [151]. The immune compartment will provide extrinsic levers. For example, CRISPR-mediated modulation of complement pathways (e.g., C3 and C5aR1) resulted in reduced secondary loss of tissue, while editing receptors on microglia and macrophages (e.g., TREM2, MerTK) enhanced the clearance uptake of myelin fragments and debris for extended periods while decreasing chronic inflammatory ‘niche’ regions [152]. With the use of AI to analyze spatial transcriptomics, we see distinct differences in immune phenotypes from the pith of the lesion to the perilesional spaces. This suggests that our editing strategies should involve regional specificity [153].

### 5.2. AI-Driven Stratification, Digital Twins, and Biomarker-Guided Feedback

For a long time now, the heterogeneous clinical outcomes with SCI have left us limited as to the generalizability of our trials. The developments in deep learning applied to diffusion tensor imaging, quantitative susceptibility mapping, and high-resolution electrophysiology offer the opportunity to cleave patient recovery into distinct trajectories. Furthermore, we are developing prognostic resolution far beyond anything our classical scales have ever afforded with serum biomarkers: neurofilament light chain, microRNAs, and exosomal cargo [154]. With the most recent emergence of a rationale for digital twins, the potential will be even more dynamic. Digital twin technologies can incorporate lesion anatomy, vascular integrity, immune signatures, and rehabilitation considerations to generate computational avatars with individualized consideration and simulate the infinite possibilities of CRISPR editing strategies in silico [155]. Some or maybe many models will suggest, for example, early PTEN deletion with later STAT3 activation has the best opportunity for maximum axonal sprouting with less spasticity. Where these offer the prospect of predicting clinical outcomes, others will offer the opportunity to quantify uncertainty and alert clinicians when biological variability renders predictions meaningless [156].

Liquid biopsies provide a “dynamic” feedback loop. In the blood we have circulating microRNAs, which have been tested and shown to relate to axonal sprouting, and the turnover of cells will no doubt be indicative with cfDNA fragments, as well as measurement of proteins as exosomal cargo. AI models could incorporate circulating miRNA and cfDNA as orderable surrogate molecular readouts of regeneration progression, supporting a minimally invasive, ongoing backdoor for regeneration monitoring. This track timing of editing interventions to biologically driven signals is more than random timing.

### 5.3. Biomaterials and Co-Optimization of Scaffolds with Edited Cells

Lastly, we have the physical lesion space, which may be the most daunting barrier. Biomaterial scaffolds (fibrin hydrogels, electrospun nanofibers, and 3D printed conduits) support structural bridges and trophic compartments. Generative AI models will allow us to develop scaffold architectures that incorporate porosity to allow for vascular ingrowth but also to orient the scaffolds horizontally to allow for axonal guidance [157]. In addition, these projections can also be validated in silico using the predicted sprouting and diffusion behaviors, and we will be able to bypass most of the iterative fabrication [158]. The CRISPR system will work to positively advance scaffold integration at the cellular level. For example, grafted stem cells with induced expression of integrins show better adhesion; modified hypoxic response pathways protect cells better from cell death in ischemic cores. When we multiplex edit cells’ properties, there is the potential to elicit cellular or tissue-level responses to scaffold gradients to rigidity or metabolism, where only after certain threshold points are exceeded do trophic factors or matrix-degrading enzymes begin to be released [159]. The AI will also co-optimize the scaffold architectures with the programming of the cells, providing designs that are each optimized collectively and not separately, enabling us to produce dynamically changing synthetic-biological composites to evolve with regeneration [160].

### 5.4. Rehabilitation Robotics and AI as Functional Sensors

We learn function and not histology. Robotic exoskeletons, neuromodulation platforms, and brain-spine interfaces can enact structured rehabilitation, but they are also diagnostic probes to extend the explanatory constructs that underpin neurophysiological states. These devices enable us to observe for long time periods where kinematics, force dynamics, and electromyography streams can be continuously recorded to synthesize extremely rich datasets that can be linked to contrived technical elements [161]. In this context it will be vital to know if the neural remappings engendered by CRISPR-based interventions were associated with clinically meaningful functional outcomes or maladaptive recruitment circuits developed [162].

Additionally, the rehabilitation robotics can also play a part beyond measurement roles by participating in adaptive protocols. For example, AI can adjust the resistance of the exoskeleton or the modulations of stimulation in real time to bias plasticity. In this case it will also be important to check if the cellular or compositional changes instigated by the molecular edits allowed the patients to capitalize on these changes [163]. This reconceptualization presents rehabilitation as an active experimental platform that allows for the functional expression of editing to be tested while measuring the intervention therapeutically and behaviorally [164].

### 5.5. Closed-Loop Regenerative Ecosystems

At the highest level of development is coupling endogenous system(s) in which the interventions are continuously updateable as biological and functional signals emerge. For example, biosensors could quantify electrophyiologic activity, spinal field potentials, and gait patterns, and capture biologic biomarkers that are detectable. These would be feeding data to a multimodal pipeline capable of using AI to classify the states of either adaptive or maladaptive [165]. When signatures of maladaptive plasticity are detected, for example, spasticity, neuropathic pain, and/or epileptiform activity, the AI could offer the following interventions from the countermeasures; indices for implementing CRISPR editing for modifying excitatory–inhibitory balance (e.g., GAD1, solutions from the nullioligan isoforms), recalibrate to the scaffold used for the rehabilitation, and change the types and pattern delivery of stimulation [166]. While not integrated in a “loop” per se, AI models were able to recommend CRISPR perturbations in spinal organoids that provided transcriptomic and electrophysiologic outputs that case by case improved the model. The discovery was expedited while scaling an essence and scale of adaptive interventions based on an individual’s biologic state. From a clinical perspective, it is conceivable that these loops might integrate digital twins, liquid biopsies, subcutaneous wearable monitors, and neuromodulation therapies into closed loops, whereby the therapies evolve in response to biology instead of fixed protocols [167].

### 5.6. Cross-Disease Insights and Broader Implications

SCI is rapidly being reconceptualized as a model for regenerative medicine. For instance, stabilizing any vasculature may be relevant, if not consequential, for ischemic acute stroke, for if the CRISPR-AI strategies support stabilizing endothelial junctions and risk is assessed for hemorrhage risk. The reconceptualizations associated with scar modulation may provide information for practice for a traumatic brain injury (TBI), considering the astrocytic hypertrophy that mediates plasticity [168]. Interventions targeted at immune reprogramming or re-education would offer close rationale within clinical care targeted for individuals with multiple sclerosis, noting that much was being developed due to the distinctions and differences in function based on the variability in phenotypes of both microglia and macrophages that modulate/remediate demyelination and remyelination. Similarly, neurodegenerative disease systemic limits and limitations share signatures of capacity/signature severity and degradation rates sensing; for example, in metabolic, perfusion, and inflammatory constraints have potential opportunities presented as frameworks developed for SCI using thinking about engagement for potential pathways, creating and sustaining an evolving state [169]. By re-contextualizing scarring and spinal cord repair, as systems of regeneration and repair there remain the potential implications for considerable engagement for a range of experiences. The possibilities could appear in the corollaries of relevance across clinical neuroscience—regeneration at a distance [170].

As we have emphasized the discussions of spinal cord regeneration content, we will conclude with one theoretically based example of how barriers we viewed as impermissible have been reconceptualized as leverage points to change and edit, connecting for the potential of steering towards a linear or predictive path of integration. CRISPR provides a potentially applicable molecular way to reprogram and/or sustain the complexity of neurons, glia, vascular, and immune niches; AI allows multimodal datasets to be integrated in a manner that stratifies possible patient groupings and generates risk trajectories and outcome predictions; biomaterials provide cost demand co-optimized for structure; robotics maintain the sensor and rehabilitation function; liquid biopsies provide molecular readouts; and closed-loop systems provide an evolutionary cycle of response [171]. Collectively, these components serve as a broad framing for revisiting repair of SCI, from an array of disconnected systemic interventions to a learning emfand ecosystem that is adaptive and translates to neuroregeneration, systemic and potentially translatable engagement. While the logistics of maneuvering and safety and ethics–landscapes for bundling these frameworks will continuously demand our collective engaged attention, we can reasonably posit that precision-guided spinal cord regeneration represents not an isolated modal alternative but also consideration for adaptive and regenerative possibilities for the CNS as a whole [171].

## 6. Convergence in Adaptive Neuro-Oncology

GBM embodies the ultimate limit of human disease adaptability: a malignancy woven from plasticity into malignant persistence, where therapy alters tumor states rather than eliminating those states, and the microenvironment is changed as much to limit repair as to promote survival. In GBM we are looking at a GBM state, in contrast to SCI, where we trial and attempt to activate regeneration in the presence of inhibition; for GBM we are neither allowed nor can inject stem cells or other progenitor cells—the paradox rests on uncontrolled regeneration as a system evolving without selective pressure [172]. Recurrence is nearly universal no matter what level of resection has occurred or treatment has been applied, such as radiotherapy and alkylating chemotherapy—recurrence is rooted in heterogeneous lesion makeup, metabolic reprogramming, nested/synaptic re-integration with host neurons, and extreme immunosuppression [173]. This coupling means that the fusion of CRISPR-based genome and epigenome editing with AI-enabled predictive, adaptive, and uncertainty-smart analytics reframes those therapies to target not fixed lesions but anticipating and listening for pre-emptive action to counteract evolution [114].

### 6.1. Mapping Heterogeneity and Evolutionary Trajectories

Mapping heterogeneity and evolutionary trajectories single-cell multi-omics suggests that GBMs exist as a heterogenous mixture of diverse, yet plastic, cellular states or conditions, such as some of a neural progenitor-like state, oligodendrocytic precursor-like state, astrocytic state, mesenchymal state, etc. [174]. Critically, their interdependence is most important to appreciate. When we think about the experimental therapy-induced transitions of neural progenitor-like cells to resistant mesenchymal fates mediated through radiation therapy, these transitions drive recurrence [175]. AI-powered models of single-cell and spatial transcriptomics data allow us to now describe state changes as trajectories along subtle high-dimensional landscapes and to predict their likely evolution under therapeutic pressure [176].

CRISPR-based pooled perturbation screening provides causal resolution to those trajectories and is able to identify “switch” genes that are biased towards the fate process of resistance. Our capability to combine MT-BIO perturbation data with AI-based trajectory models enables predictive mapping of heterogeneity, termed “heterogeneity guidance”, where therapies are contingent not just on current nearest neighbor composition but on predictions on the clonal future [177]. Lineage barcoding and CRISPR-mediated editing allow for dynamic tracking of clonal branches over time, validating predictions as well as providing labeled training data that inform AI systems. This integration re-conceptualizes heterogeneity as a dynamic system that can be modeled and perturbed with the potential for guidance versus an insurmountable obstacle [178].

### 6.2. CRISPR Reprogramming of Tumor States

#### 6.2.1. Oncogenic Drivers and Synthetic Vulnerabilities

CRISPR-mediated disruption of recurring GBM oncogenic drivers (EGFRvIII amplification, IDH1-R132H mutation, CDKN2A deletion, and TERT promoter activation) is still an essential first step. Multiplex CRISPR constructs allow for the concurrent disruption of redundant oncogenic circuits, minimizing the chances of clonal escape. AI-supported prioritization of synthetic lethal pairings can ensure that edits are state-specific, exploiting vulnerabilities that are not those of the mesenchymal- or progenitor-like clones [179].

#### 6.2.2. Epigenomic Stabilization

The heterogeneous adaptations of GBM rely on epigenomic plasticity. CRISPRi and CRISPRa platforms, coupled with chromatin remodelers, can provide modification of enhancer-promoter networks, precluding any transcriptional transitions in cells that might advance into resistant states. Editing oxygen-regulated chromatin regulators extends this thinking into hypoxic niches wherein an excess of metabolic stress has changed the preservation of metastable epigenetic programs [180].

#### 6.2.3. Therapy Sensitization

Making resistant tumors into therapy-sensitive tumors is possible using CRISPR. Altering MGMT promoter methylation restores temozolomide sensitivity, and disrupting the essential regulators of homologous recombination augments radiotherapy. These anaerobic and metabolic pathways allow for the augmenting of precision editing with pre-existing therapeutic pathways, creating fewer barriers to clinical translation [181].

#### 6.2.4. Immunoediting

The entirety of the immune microenvironment in GBM is a span of the expression of checkpoint ligands (PD-L1), suppressive cytokines (TGF-β), and the recruitment of regulatory T cells and myeloid-derived suppressor cells. CRISPR has the power to silence PD-L1 or reprogram tumor cytokine profiles when engineering CAR-T or NK cells to minimize exhaustion and enhance trafficking. AI-enabled neoantigen identification increases the utility to prioritize new epitopes for CRISPR-augmented immune targeting and trigger the evolution of immune pressure notwithstanding tumor adaptation [182].

### 6.3. Tumor Ecology: Vascular, Metabolic, and Neuronal Niches

GBMs do not just grow by autonomous mutations but via their ecological niches. Through pathways delivered by chemokines and through hypoxia stabilizing HIF1α and HIF2α to drive angiogenesis through VEGFA and ANGPT2, GBMs access and colonize angiogenic ecological niches [183]. For example, tumor cells can bypass hostile hypoxic niches through vascular mimicry via trans-differentiation; re-programmed tumor cells actively express an endothelial-like state. AI, with perfusion MRI, hypoxia PET tracing, and spatial transcriptomics, will provide in vivo mapping of ecological niches and facilitate stratified spatial CRISPR-based interventions [184]. Metabolic edits targeting lactate shuttling (e.g., MCT1/4), glutamine metabolism, or targeting cholesterol could shut down the tenuous nascent metabolic landscape wherein resistant clones emerge and adapt/survive through biological selection [185]. Equally novel is the coupling of GBM to neuronal circuits through synapse-like connections. AMPA-mediated synaptic interfacing with adjacent neurons magnifies the GBM’s proliferative and invasive potential, which is somewhat suppressed by activity-dependent neuronal release of neuroligin-3; therefore, CRISPR targeting neuroligin-3 signaling, AMPA subunits, or downstream calcium effectors affords the chance to ‘decouple’ GBMs from the neuronal networks they exploit to drive their growth. If AI-assisted EEG and connectomics studies ultimately make it possible to anticipate that neuronal activity is eclipsing to trigger invasion and allow treatments to be administered at the right time [186].

### 6.4. Delivery: Engineering Access to the Brain

CRIPSR-based neuro-oncology is based on delivery across the blood–brain barrier and infiltrative margins. AI-driven design of nanoparticles has enabled lipid formulations containing transferrin or RGD ligands to preferentially accumulate in GBM tissues. For example, focused ultrasound (FUS) with microbubbles is a non-invasive modality that can temporarily disrupt the BBB. AI models can optimize the sonication parameters in real-time to maximize penetration potential and ensure a safe delivery is achieved [187]. While convection-enhanced delivery (CED) has existed for decades as an experimental platform, it is now being applied in a new use with experimental use again. The simulation of fluid dynamics using AI allows for exciting new adaptive delivery possibilities with CED, such as potentially dynamic changes to the delivery vector of high-value agents (e.g., Cas components or NPs) using real-time CAT scan data, real-time catheter placement, and infusion rate [188].

Exosome-based carriers present a similarly exciting platform. Tumor-homing exosomes designed to load with Cas components have inherent sequential screening abilities that, combined with biocompatibility, offer a tremendous drug delivery platform. AI models predicting exosome sorting mechanisms and biodistribution are developing the rationality behind vesicular vectors [189]. Likewise, viral platforms have remained underutilized due to immunogenicity, but AI modeling of directed evolution of capsids is happening, which can enhance tropism and contain off-target spread. Together each of these exploitable platforms constitutes a new foundation of delivery as a non-static barrier to modifiable, tunable variables within adaptive neuro-oncology [21].

### 6.5. Adaptive AI–CRISPR Feedback Systems

For full chow adaption to take place, constant closed-loop monitoring needs to commence. Exciting advances in the development of circulating tumor DNA, CSF-derived exosomal RNA, and serial imaging are providing very rich temporal “biomarkers” of neoplastic dynamics. With the dimensional data richness multimodal data can provide, AI systems will be able to adopt models to deal with the different data streams and detect early signs of clonal expansion or therapeutic escape. This will lead to changes in CRISPR strategies to target pathways for editing emerging clones, changing immunotherapy targets, or forecasting metabolic dependencies [190].

Delta-radiomics—AI-based reading of imaging change vs. static imaging—is developing our sensitivity to detect early adaptive turns. Specifically, when combined with intraoperative rapid single-cell sequencing and mass spectrometry, our approach is also able to afford real-time shifts to therapy in the time sequence of intraoperative therapy delivery [191]. Rather than intraoperative trial-to-trial, the organoid-in-the-loop model extends this concept ex vivo: patient-derived GBM organoids would be CRISPR perturbed; AI would interpret the phenotypic responses, and then the interpreted data would inform the individualized therapy for the patient. This model could enable us to conceptualize cancer based on an adaptive learning model that aligns with the adaptive biology of evolving GBM [192].

### 6.6. Toward the Adaptive Operating Room

A long-term vision places this adaptive loop in the operative setting. In trauma to the patient, the new resected tissue will not only undergo near real-time sequencing but also CRISPR perturbations; the AI pipelines will signal resistance trajectories, drug vulnerability scoring is generated before any saves, and the immune cells, or molecular edits, are developed in the surgical context to be re-infused post resection [97]. Moreover, visualizing the engineered immune or molecular targets in our neuro-navigation system, we eliminate the subjectivity out of navigation, as the navigation relates to the patient’s anatomy, thus reconceptualizing the OR as a computational space where the evolution of tumors supports the continual adaptive learning. In neuro-oncology, this conceptual incubation does not provide equipment and talent on episodic interventions but as a continually adaptive learning system based in the specific clinical context of the practitioner [193].

### 6.7. Risks, Safeguards, and Trial Design

There exists an unknown risk with CRISPR: off-target edits, mosaicism, and the potential to rewrite the cellular pathways of the tumor. In relation to AI systems, these series can be multidimensionally biased, misclassified, or overfit. Furthermore, there can be tumor acceleration (or paradoxical progression) if the edits, or perturbations incompletely inhibit existing adaptive pathways [194]. To protect the potential risks of off-target edits off target edits can be monitored with GUIDE-seq, and CIRCLE-seq with long read sequencing, which would report unintentional changes when CRISPR DNA double-strand breaking occurs along with structural variants, and predictive off-target models could also be correlated using chromatin features. In addition, combining immune therapies with suicide switches and logic-gated CAR constructs targets the acute immune response to the efficacious but neurotoxic suppression of GLIA infiltrating tumors [195].

Translational research objectives require adaptive trial designs. One possibility is that a BC Bayesian multi-arm trial platform would include multiple members and allow multiple CRISPR–AI experiences to run at the same time, where the trial arms evaluate molecular biomarkers for pragmatic endpoints rather than an accurate accounting of limits of patients mobility in clinic after being recorded as having had progressive disease [196]. Digital twins (computational avatars capture the benefit of multi-omics, imaging, and biosignals from clinicians that provide for a unique simulation experience for each patient for dosing/eligibility support) as federated learning allows each center to update individual models while each center retains decentralized data at their central node. These models ought to position the ideas of identity and innovative ethical and regulatory anticipation while keeping accountable those overseeing the balance of all adaptive approaches [197]. The necessity of requiring novel therapies that are adaptive in time to the unrelenting evolution of GBM and its microenvironment has been directly drawn from this discussion. By capitalizing on CRISPR, we have tools that can shut down oncogenic drivers, keep the epigenomes biologically chronically stabilized, induce tumor cells to ante up and sensitize themselves to existing therapies, and modify one or more immune, vascular, metabolic, and neuronal niches of heterogenous GBM [198]. We now possess AI-equipped systems frameworks that are capable of describing heterogeneity and possible evolutionary trajectories while allowing the inclusion of any biomarkers individually across time and real-time spatially, and engineered biotechnologies in concert with drug delivery in vivo to the natural systems. Furthermore, all of this new technology has moved towards a model of adaptive neuro-oncology that leads to timely real-time interventions made ex vivo and being able to track closely in situ tumor evolution at the same surgical operation timespace [199]. Furthermore, the laboratory and operative spaces could integrate their kernel infrastructures into molecular and computational hybrid growth. By knitting unrelated strands together, that collectively will augment the evolution of therapeutic evolution in the GBM paradigm—GBM reoriented as a staged, modifiable system rather than an existing rule of recurrence. All of this could ultimately provide conceptual support to the design of precision therapies that evolve along with the malignancy being treated and the treatment of the malignancy.

## 7. Expanding Horizons: Beyond Spinal Cord Injury and GBM

Although SCI and GBM are exemplary models of regenerative challenge and malignant adaptation, the intersection of CRISPR technologies and artificial intelligence extends far beyond these two examples. Stroke, neurodegenerative disorders such as Alzheimer’s and Parkinson’s disease, rare neurogenetic disorders, and TBI are additional domains in which molecular programmability and computational predictions can be incorporated into therapeutic modalities [162]. The following subsections will attempt to provide insight about each of these conditions regarding molecular precision, delivery, AI scaffolds, and ethical predictions, while providing examples of how these disparate conditions pull together into an integrated regenerative logic.

### 7.1. Stroke and Ischemic Injury: Digital Penumbra and Network Reconnection

Ischemic stroke reflects a rapidly evolving chain reaction: vascular occlusion results in ischemia, and a cascade takes off with excitotoxicity, mitochondrial dysfunction, oxidative stress, BBB breakdown, and immune infiltration as each node in the chain is stimulated. Emerging CRISPR interventions are specifically designed to manipulate each node of the ischemic stroke pathological cascade [200]. For example, editing the tight-junction components of claudin-5, occludin, ZO-1, and VE-cadherin stabilizes tight-junction integrity and limits paracellular leakage and secondary edema. The communication between astrocytes and endothelial cells can be modulated by altering PDGFRβ in pericytes and signaling in MC barrier endothelial cells via S1PR1–eNOS, promoting stability at the capillary level [201,202].

At the neuronal level, CRISPR knockdown of GRIN2B or disrupting the DAPK1–NR2B interaction decreases NMDA receptor-mediated excitotoxicity. Meanwhile, modifying the chloride homeostasis by upregulating KCC2 and downregulating NKCC1 provides inhibitory balance to peri-infarct neurons. In the same way, prime editing mitochondrial regulators such as PGC-1α and TFAM, or enhancing mitophagy by either PINK1 or Parkin, can maintain bioenergetics. Alternatively, pro-survival pathways (Akt, STAT3, BCL2) might be increased transiently by CRISPRa to delay crossing the threshold for apoptosis in these neurons [203]. Regeneration is about more than just cell survival. Nevertheless, CRISPRa could modify astrocytes and NG2-glia to neurons via transcription factor (NeuroD1, Ascl1, Sox2) modulation, and oligodendrocyte maturational factors (SOX10, OLIG2, MYRF) can activate remyelination; the transition would restore synaptic and axonal repair [204].

AI models may behave like an evolvable intelligence. Digital penumbra twins generate the perfusion/diffusion MRI, ultra-fast ultrasound, EEG, and biomarker profiles (NfL, GFAP, and microRNAs) and are yielding real-time estimates of salvageable tissue as the diagnostic profiles yield stratification and eligibility for treatment, predicting risk edemas/hemorrhage [205]. AI models provide the foundation to rank spatial targets to complete CRISPR edits and those schedules for the biomolecular therapy in the order of sequencing—establishing barriers→ counteracting excitotoxic injury→ stabilizing mitochondria→ repositioning. Functionality observational outcomes by robotic rehabilitation device/BCIs provide the adaptive feedback loop indicating the biochemical changes were made, and functional circuitry has been re-established [206].

### 7.2. Neurodegenerative Diseases: Engineering Resilience in Progressive Collapse

#### 7.2.1. Alzheimer’s Disease

Alzheimer’s disease is typified by amyloidogenic processes, tau hyperphosphorylation, synaptic failure, and microglial remodeling. CRISPR technology can provide laser-like precision entry: silence BACE1 and prevent generation of amyloid; edit γ-secretase subunits, which may alter substrate specificity; and mitigate off-target effects. Tau-related pathology may be targeted with CRISPRi that targets endogenous MAPT splicing regulatory sequences that interact with other targets to demote the accumulation of prion-like isoforms [207]. Compromised competence of microglia could potentially be addressed by targeting TREM2, TYROBP, and CD33 to modify synaptic functional fatigue via modification of complement (C1q, C3, C5aR1). Notably, base editing of the APOE alleles may be a game-changer: re-specifying ε4 variants to act more like ε2—poor lipid transport of amyloid. Also, changes in astrocytic lipid metabolism have been shown to occur via ABCA1/LXR modulation for targeted resilience specifically at the synapse. The role of AI supports all of these intervention frameworks by permitting early staging using plasma p-tau217, Aβ42/40 ratio, NfL, multiple imaging modalities (amyloid/tau PET and fMRI), and digital behavior phenotyping [207]. Predictive algorithms stratify patients in relation to the course of disease, so CRISPR can be delivered preclinically before permanent loss of neuronal networks. Adaptive AI allows for early real time ongoing assessment of biology and signals obtained prior to CRISPR delivery, and the timing of CRISPR delivery can be adjusted based on disease phase transitions that occur over time [208].

#### 7.2.2. Parkinson’s Disease

Parkinson’s disease exists in biogeochemical clusters of α-synuclein aggregation, mitochondrial dysfunction, lysosomal overload, and associated degeneration of dopaminergic cells. CRISPR strategies to date have focused on silencing SNCA to reduce aggregation, silencing LRRK2 and GBA1 mutations, and enhancing mitophagy, primarily with PINK1/Parkin. CRISPRa to activate transcription factors (NURR1, LMX1A, and FOXA2) can allow the development of dopaminergic neurons from astrocytes for localized cellular replacement [209].

Integrating AI allows for streaming multimodal data from wearables, gait trackers, speech analytics, and telemetry from deep brain stimulators into individualized circuit maps. These circuit maps can act as two repair paradigms at once; molecular edits will replenish depleted neuronal populations, and adaptive DBS will align basal ganglia circuit maps with contemporaneous newly developed dopaminergic neurons. AI guided co-adaptive controllers will regulate stimulation envelopes to deliver consistent stimulation as circuits are regenerated and mature. This maintains stability and reduces dyskinesia or maladaptive plasticity [210,211].

#### 7.2.3. Rare Neurogenetic Disorders: Trajectory Aware Allele Correction

Monogenic disorders require exactitude at the appropriate level. For instance, with Rett syndrome, allele-specific restoration of MECP2 through CRISPR will mediate dosage sensitivity. Base editing of SCN1A in inhibitory interneurons for Dravet syndrome will correct E/I balance, thereby limiting seizure burden. In Huntington’s disease, peak editing to shorten the number of CAG repeats on the HTT gene or using allele-selective CRISPRi to silence expression of the mutant transcript while preserving the wild type transcript can be seen as a precision approach to the problem of the disease [212]. Similarly, as an RNA-targeting CRISPR, RNA-targeting Cas13 can be useful when used for transient inhibition of toxic repeat RNAs and counteracting dipeptide repeat toxicity. In 2024–2025, gene-writing CRISPR–transposase fusions will provide the capacity for large-scale restructuring of expansions and other structural variants, taking editing into arenas inaccessible to traditional nucleases [213].

AI is harnessing these applications via interpreting the pathogenesis of variants with conservation, prediction of splice sites, and context of expression. In pediatric digital twins, genomics, connectomics, developmental trajectories, and electrophysiology enable forward modeling of outcomes based on variants and diverse editing approaches. By mapping intervention windows based on developmental milestones, AI facilitates the construction of trajectory-aware allele correction where molecular editing occurs in coordination with neuro-developmental plasticity [214].

### 7.3. Traumatic Brain Injury: Closing the Loop on Inflammation and Repair

Unlike targeted ablative injuries resulting from a gunshot to the head, TBI has an acute mechanical injury process that includes microvascular disruption, ionic dysregulation, oxidative damage, and, in many cases, chronic neuroinflammation. Greater understanding of acute and chronic inflammatory injuries has created opportunities for CRISPR to mitigate persistent inflammatory injury by targeting NLRP3, IL-1β, TNFα, C3, and C5aR1 to promote recovery, as well as enhancing antioxidant stress defenses through SOD2 and NRF2 editing, along with a restoration of levels of BDNF, NGF, and STAT3 to promote a regenerative and resilient neural response. The astrocytic-to-neuronal reprogramming mechanism can replace inhibitory interneurons to lessen post-traumatic epilepsy, while oligodendrocytes can be supported for limb regeneration through MYRF induction for remyelination possibilities [215,216]. AI holds and integrates multimodal markers, meaning using the biomechanics of injury, imaging with MRI and DTI to measure tract integrity, susceptibility-weighted imaging for observing microbleeds, EEG as a predictor of seizure risk, and speech and oculomotor markers for neurocognitive decline. Machine-based approaches to learning the chronic outcomes risk stratifying patients and predicting maladaptive plasticity and chronic outcomes such as spasticity or neuropathic pain [217,218]. It is at the interface of inflammation and CBA/CRISPR, or the adaptive controller aspect of TRIP/CRISPR, in collaboration with a helpful human informatician who can time CRISPR interventions, pausing or altering the planned milieu. The net product is a closed-loop cycle of immune-neuroregeneration where immune-modulation and regenerative repair support each other [219].

### 7.4. Cross-System Interfaces: Endothelial, Immune, and Metabolic Crosstalk

Neuropathology exists in a conjunctive space and not solely in neurological symptoms; thus, the vascular, immune, and metabolic systems must co-localize and configure together at all phases of CNS repair. Modulation of the immune system suppressed remote IL-6/JAK/STAT3-mediated delayed secondary damage in a system of stroke or TBI. In AD and PD, modulation of immune responses of lipid and glucose metabolism (APOE, ABCA1, PGC-1α) has been shown to support neuronal viability [220]. The peripheral mitochondrial regulators are found in the liver and skeletal muscle, which then act as metabolic counterparts of CNS repair; CRISPR-based modulation of these same nodal locations has produced stackable, systemic neuro-regeneration [221]. AI will align the pathway bioinertia for the connections mapped to assess both central and peripheral markers; likewise, also ensure peripheral knots at the node do not disallow global CNS repair [222].

### 7.5. Delivery Innovations Across Disorders

Delivery systems remain a limiting knot in the system but have made great inroads. FUS with micro-bubbles creates the possibility of transient permeability of the BBB and possible regional dosing in AD and stroke or diffuse TBI. AI has resolved priors on dosing packets, transferring safety protocols for FUS sonications. CED should also be enhanced with AI flow modeling, developing likely therapeutic distributions through the more likely different tissue micro-environments [223]. Dosing of intranasal delivery systems is also only becoming more engineered, with nanoparticles enabling non-invasive repeated outpatient dosing in chronic indications. AI-trainee exosomes with known surface signatures are now promising for cellularly selective targeting of CRISPR cargo delivery and fine-tuning control of gene expression [224]. Hydrogel depots of CRISPR-exosomes make it possible to deliver localized or continued release of CRISPR cargo into the brain, either in cortical or hippocampal areas in sequence. Darwinian selection of viral capsid evolution could also be directed from advanced generative AI-styled models of migration and tropism to appropriate neuronal populations at risk- cortical layer II/III neurons at risk during AD or nigral dopaminergic neurons at risk during PD [225].

### 7.6. Ethical, Translational, and Trial Considerations

Precision genome editing interventions already present ethical dilemmas. Minimal-to-absent off-target edits, structural variants, and mosaicism will necessitate orthogonal detection systems (GUIDE-seq, CIRCLE-seq, long-read sequencing). In the case of pediatric disorders, decision-making about when genomic edits take place, preceding symptom onset, continues as an entangled novelty that complicates consent for and certainty regarding the genetic assessments [226]. When the established genomic edits unoccur to a population of APOE alleles, using the population evidence presents obvious and unsolved philosophical dilemmas for questions of quality and equity of availability of informed access indexes, availability of other domains for equitable neuro-enhancement, and inserting a TBI presenting problems for first responders with conflict for dual-use in a sport action that is not diabolically opposed to military action [227].

AI will yield ethical dilemmas, in particular algorithmic bias, models, replicability, and ideas of reproducibility in clinical trial reporting where accountability remains a question. Governance mechanisms will need to consist of temporal and version changes to its models, each with audit trails and publicly available, added with human-in-the-loop oversight [228]. The feasibility of an adapted Bayesian trial design that applies molecular surrogates for improving efficiency to translation could be possible within a federated analytics hub while also assuring privacy. In theory, digital twins might obstruct the feasibility of identification for patient eligibility and, thus, for dosing adjustment and/or safety limits required for delivering meeting ethical thresholds [229].

### 7.7. Toward a Universal Regenerative Logic

Historically, and in all the described circumstances of preconditions, intervention modalities, and domains of multiple contextual levels; it has been intriguing that we observed and abstracted patterns of experiences and thematics: vascular stabilization, mitochondrial resiliency, immune rebalancing, synaptic support, neuronal replacement, and rail signaling for lingering, at least adding integrity to the cerebellar circuitry [230]. By the time CRISPR programs are tuned with molecular levels, we are tasked with applying AI as manager of timing, spatial targeting, and adaptive real-time monitoring of whatever goes unresolved. Stroke has shown the penumbral connectome, PD positively illustrated the dual repair paradigm; rare disorders like restore presented with allure for the possibility of trajectory aware allele correction, and TBI focused on doing the closed-loop immune–regeneration cycle acknowledged a responsive systems cycle using its totality of principles of the modality of global implemental quality improvement as a strategy to completion. Collectively, these funeral frames of reference only intended to coalesce, converge, into a universal regenerative logic—either uniquely random or freshly observed neuro-repairs that adhere to precision editing from a random, ghostly remnant outcome or rare occurrence toward a well-behaved, predictable, and transparent process of functional repairing.

## 8. Building the CRISPR–AI Neurotherapeutic Pipeline

The evolution from experimental proof-of-concept to routine clinical practice will require a therapeutic pipeline that is not linear but recursive, adaptive, and with high assurance. A system of this caliber captures and harmonizes multimodal data types, integrates and interprets these multimodal data types through computational models, designs and deploys molecular interventions with proven safety, incorporates a phase of continuous monitoring to close the loop, and builds to an approach that is not linear in sequence but a platform with living architecture that learns from each patient, updates the internal model, and improves toward better precision, safety, and equality [231].

### 8.1. Input Layer: Standardized, Multimodal, and Context-Rich Data

The input phase is collecting the most broadly sourced data ever collected in neuroregeneration: various genomes, epigenomes, single-cell transcriptomes, proteomes, and metabolomes; spatially resolved tissue maps; quantitative MRI, DTI, and PET; electrophysiological streams from EEG, MEG, and/or implanted electrodes; as well as digital phenotyping from wearables and/or robotics [232]. The difference the next generation of inputs affords is not just a data mountain, but standardization. Neuroregenerative CRISPR-AI pipelines to be reproducible and comparable across sites require data stores to comply with metadata standards, ontologies of cellular states and microenvironments, as well as harmonized acquisition protocols. New initiatives such as a Regenerative Data Commons will posit that all datasets (organoid models, animal systems, and human patients) be deposited within interoperable frameworks similar to GA4GH standards in genomics to advance regenerative health [233].

This input phase will be complemented by liquid biopsies (cfDNA, exosomal RNAs from peripheral circulation, circulating proteins, etc.), which are minimally invasive molecular readouts, and will be supplemented by ecological monitoring (wearable sensors and speech and motor function analyses), which will provide continuous functional phenotypes. Together, these data sources do not provide simply a static atlas but rather a temporally resolved landscape of neural states that computational interpretation can draw upon as raw substrate [234].

### 8.2. Processing Layer: AI Integration with Causality, Control, and Verification

Once the inputs are captured, they are processed through AI models that not only identify correlations but also infer causality and provide transparent, actionable outputs. Cross-modal transformers and graph neural networks relate omics with imaging and electrophysiological measurements into shared latent spaces, and topological methods have been developed to identify uncommon or transitional states, which are often critical for regeneration or oncogenic transformation [235]. The defining aspect of this layer is the grounding in causality and hence counterfactual reasoning. The models are generalized to incorporate biological priors and perturbation datasets to predict what is associated with repair as well as what would change if a specific CRISPR perturbation were to occur. Active learning strategies are utilized to inform the next most informative experiment; the latter could be multiplex editing an organoid so that the laboratory experiments would be driven by model uncertainty and not just by intuition [236]. The importance of control theory and formal verification cannot be overstated. Treating the pipeline as a cybernetic loop with sensors (biomarkers), controllers (AI models), actuators (CRISPR perturbations), and safety interlocks (rollback mechanisms) will allow us to leverage rigorous engineering principles based on aerospace and nuclear systems to enforce constraint specifications (e.g., maximum permitted risk, bounded error rates). In addition to conveying the recommended perturbational experiment, the models convey the model’s calibrated uncertainty, counterfactuals, and an audio trail of training data to ensure transparency and reproducibility. The output of the pipeline is the conversion of computational predictions into actual molecular interventions. Guides are designed against the genomes of the individual, as opposed to a reference genome, which allows for pangenomic variation and allele-specific variations to be accounted for. This permits allele-selective editing in disorders such as Huntington’s disease or in cases of Alzheimer’s disease modified by APOE. The choice of editing platform—nuclease, base editor, prime editor, epigenome editor, or an RNA-targeting system—is based on the sequence context, cell-cycle state, and therapeutic requirements [237].

Delivery is optimized with AI-guided capsid engineering due to the need to contact and deliver edits to target cell types while using convection-enhanced infusion models to keep off-target exposure to a minimum. Expression of modified genes or cell death is regulated using cell-type—specific promoters or microRNA gates, and logic-based control circuits provide further levels of safety. For example, we can develop split nucleases that reconstitute only in the specific environment, build small-molecule switches to either activate or deactivate circuits, or even design suicide modules that terminate the therapeutic intervention if an adverse event arises. The logic-enabled philosophical shift provides the potential to move CRISPR from a blunt intervention to a programmable therapeutic device [238].

### 8.3. Feedback Layer: Continuous Telemetry and Adaptive Control

The most significant feature of this pipeline is that it is a closed-loop system. With respect to measuring any molecular edits, real time monitoring provides amplicon sequencing, long-range off-target detection, and liquid biopsy tracking of the edited loci. Functional recovery is tracked simultaneously using electrophysiology, behavioral kinematics, neuroimaging, and cognitive measurements. AI takes these feedback modalities into account using adaptive feedback policies [239]. If recovery is delayed or flat, or off-target edits occur, AI can model the gRNA design or shift cognizant therapeutic focus from axonal outgrowth to synaptic stabilization. Rollback mechanisms offer surety; anti-CRISPR proteins, vector inhibition even immune modulation all could be turned on if telemetry suggests emerging threats. Rollback mechanisms suggest an intervention is no longer permanent but an altered trajectory that is dynamically modulated to create optimized trade-offs between regeneration and stability [71].

### 8.4. Bridging In Silico, Organoids, and In Vivo Models

A new layer in the pipeline is the use of organoid twins—brain organoids that are edited with CRISPR and monitored with single-cell and electrophysiological readouts—that provide an intermediate validation layer between digital simulations and live patient in vivo models. The organoid twins provide a method to evaluate AI predictions in living tissue architectures before advancing to human translation. Equally, the microfluidic BBB-on-chip systems that have been designed by AI optimization provide the opportunity to screen delivery vehicles under plausible physiological conditions, and bridging across layers—digital, organoid, and clinical—helps to create a continuum of evidence to demonstrate an intervention is both safe and predictable [240].

### 8.5. Clinical Translation and Trial Innovation

When translating to the clinic, the pipeline will need trial designs that are accommodating of its dynamic nature. Traditional fixed-arms trials will no longer be viable; however, platform trials with shared controls and Bayesian adaptive randomization would enable simultaneous testing of multiple CRISPR–AI interventions in real time as both the predicted trajectories and response to the interventions are adjusted. Eligibility could be modeled using digital twins to ensure that only patients predicted to respond with acceptable risk are enrolled [241]. Endpoints represent a paradigm shift in this tortuous landscape, not just survival or motor scales, but molecular biomarkers, imaging signatures, and network-level electrophysiological coherence, as it all adds up to an earlier, quite specific detection of therapeutic effect. Now that endpoint models have been developed and vetted by regulatory hurdles, we can integrate post-market surveillance into the pipeline, whereby real-world data, when added to endpoint models, should translate to model enhancement, as safety and efficacy are refined by the totality of global experience [242].

### 8.6. Standards, Ontologies, and Benchmarking

The benefit of the wider applicability of the pipeline would be offloaded if we had some common standards. The development of ontologies for regenerative cell states, for microenvironmental niches, and for CRISPR types of intervention will allow harmonization of results across labs and consortia. A sort of benchmarking akin to ImageNet in the computer vision literature for CRISPR–AI in neuroregeneration, with open datasets and leaderboards, could be proposed for assessment in terms of editing prediction, off-target detection, and outcome prediction [243].

By having a set of quantitative metrics, from editing efficacy and connectome remapping data to system level metrics like pipeline latency and “rollback” efficacy, we will be able to measure progress and allow comparisons. These usage measures will sustain reproducible science in the field and will make lessons learned transferable across studies.

### 8.7. Integration with Neurotechnology and Prosthetics

Beyond what is occurring on a molecular and computational level in the pipeline, it would be beneficial to integrate into applications like brain–computer interfaces (BCIs) and adaptive neuroprosthetics. Continuous cortical imaging streams will be feasible from implanted arrays, along with when there are applied, adaptive, deep brain stimulation signals, and that will provide readings of real-time cortical network excitability, which may help with structuring CRISPR interventions towards enabling management of excitatory/inhibitory balance back to a state in a healthy region of a restored musculoskeletal system [244]. On the flip side, coordinated CRISPR cellular regeneration with prosthetic modulation can ensure that the incorporation of new neurons or circuits is functionally integrated. This bi-directional melding of molecular repair with neuroprosthetic controllers represents a new frontier for precision neuro-rehabilitation [245].

### 8.8. Global Governance, Equity, and Ethics

Introducing such a pipeline creates troubling global issues. CRISPR–AI neurotherapies could likely be more available to rich countries and exacerbate inequities, so it will be imperative to consider social equity and global governance. One idea is to establish a global neuroregeneration observatory using something like GISAID, where countries contribute their anonymized data and standards to NNArms, providing essentially everyone with access to studies, policies, and best practices across national borders. Licensing and subsidy issues can rely on an evolving tiered-access approach that provides resource-poor countries access to new therapies, if possible [246].

Ethical safeguards have multiple layers beyond access: we must consider the dual-use potential that some of these emerging technologies can also serve as weapons or capabilities for enhancement. Oversight must ensure balance between the acceleration of innovation versus the responsible stewardship of these technologies. Patient advocates and members from local communities should also be embedded into governance structures to reinforce a pipeline that is not only scientifically valid but also socially legitimate [247].

### 8.9. Emerging Horizons: Quantum, Edge, and Beyond

As well, the pipeline must include the opening of emerging paradigms of computation. Quantum-like algorithms may have the potential to model CRISPR off-target energetics and protein folding landscapes. Digital twins of DNA/RNA-based storage, which can provide models of patients’ specific long-term memory storage in molecular media. Edge AI that operates on wearables and implantables allows processing biosignals locally while only transmitting compressed status- or design-Logfiles to central systems, which supports reduced latency and potential opportunities for privacy. These technologies are not yet being implemented at scale but offer horizons of the depth of computational medicine over the next decade [248].

The CRISPR–AI neurotherapeutics pipeline now exists as a more complex but coherent architecture: standardized multimodal inputs, ACT-based and verifiable computation, precision molecular output with logic, real-time monitoring closed-loop feedback, organoid- and microfluidic-mediated bridging, adaptable and appropriate trial structures, transnational standards or benchmarking, neuroprosthetic integrations, and mechanisms of governance to address equity and accountability. All of these dimensions offer pipeline options, and providing the individual nuances illustrates not a speculative vision but a simple operationalization stylistic blueprint—flexible, auditable, and supremely scalable—for ultimately enabling the clinical practice of neuroregeneration [249].

## 9. Ethical, Safety, and Regulatory Considerations

The combination of CRISPR genome editing with AI-based decision systems in neuroregeneration represents not just a scientific milestone but an exciting, ethical, and regulatory frontier. Unlike other forms of therapeutic intervention, these will act as two layers of permanence: our biological edits, irreversible and permanent for life, and computer models, continuously changing or adapting in ways that humans cannot fully interpret. This pairing creates a dual-risk environment unlike any in medicine, where policy and governance frameworks will need to develop safety, accountability, and governance systems based on accepting complexity rather than limiting it to outdated categories [250].

### 9.1. CRISPR-Specific Safety Challenges in the Nervous System

Neural tissues face specific vulnerabilities in genome editing. Unlike peripheral organs, the brain and spinal cord (CNS) only have limited levels of regenerative capacity, and their functional capabilities result from delicately established network dynamics. Unintended and subtle genetic alterations could result in circuit wide dysfunction that manifests as seizures, loss of function, or cognitive impairment. Off-target edits have been portrayed as nucleotide substitutions, similar to those genomic alterations having been restricted to translocations, large deletions, or cryptic integrations. In the CNS, this kind of unintended alteration could disconnect partners in a synaptic network or destabilize clusters of genes needed to execute an excitatory–inhibitory balance [251]. Mosaicism adds additional challenges since incomplete (failed) or heterogeneous editing could lead to cell populations with very different fates; some help establish regenerative mechanisms, while many may disrupt neurodegenerative mechanisms or intolerably encourage cell division. We should also acknowledge the unintended consequences of reactivating intrinsic regenerative growth programs in certain contexts, like the instances of PTEN or SOCS3 knockout, since there is an argument that oncogenesis can occur in non-canonical, per-adult brain tissue where connections to gliomagenesis may predispose for such outcomes. The contradiction in trying to stimulate regeneration in a system that altogether resists growth is that the pathways that were unlocked may have made it possible to promote tumorigenesis [252].

The vectors that are used for delivery are another major category of risk. AAV vectors have been extensively studied but come with different levels of risk at which insertional mutagenesis and immune priming may happen. Lipid nanoparticles and polymeric carriers are non-integrating, but still deliver cargo to inflamed CNS tissue poorly, and biodistribution across the BBB is inconsistent. Engineered exosomes and hybrid nanocarriers hold lots of promise for greater targeting specificity, but the recent literature is only just starting to characterize their immune compatibility, biodistribution, long-term persistence, etc. The combination of neuronal vulnerability, delivery to the CNS being heterogenous, and oncogenic potential means that we must invoke safety interlocks like inducible editors, suicide switches, reversible epigenetic modifiers, anti-CRISPR proteins, etc. In the design of therapeutic constructs, these measures should be characterized similarly, as a building block of safety architecture rather than optional safety features [253].

### 9.2. Ethical Complexities in AI-Guided Neurotherapeutics

Artificial intelligence presents yet another unique category of risks that do not have comfort in molecular biology but instead exist in the realms of epistemology and governance. For example, when an AI model is used to integrate single-cell omics, imaging, and electrophysiology to recommend CRISPR edits, it does so with a predictive power that defies human reasoning. In addition, a major problem with AI models is their opacity in predicting effects, which erodes accountability. Under the law, an actor must be professional and provide adequate safety/protection in the activity for patients [254,255]. However, when these biomedical interventions are initiated, the act of providing consent becomes an act of good faith trust, not just in the biological safety aspect of each intervention, but also in the accountability of a computer/inference system. If the pathway to “do no harm” is violated in establishing trust in the computer recommendation, legal obligations to individual patients are diffused across a roiling cauldron of data custodians, model developers, implementing clinicians, and regulators, thereby presenting conceptual dilemmas that existing medical malpractice or liability frameworks may not be able to resolve [256].

We further exacerbate the problem through AI model training input bias. Most AI models created for biomedical applications are based on datasets derived from mainly high-income countries, which are typically not representative of populations with different ancestries, environments, and disease phenotypes. This creates a sort of paradox of precision—marketed as personalized therapies may be least precise—least accurate—for populations that are least represented in biomedical AI and exacerbate existing health inequities [257]. Synthetic data augmentation is being increasingly deployed by researchers to ameliorate these facets of the problem, opening up new ethical dilemmas around authenticity and consent—if a patient’s digital twin is partly derived from generated data, what does this imply for ownership of, or accountability for, the digital twin? Digital twins also raise ethical boundaries that, to our knowledge, have rarely been encountered in medical practice; these computing surrogates could certainly change therapeutic decision-making before any in vivo intervention has been made. However, digital twins are not objective recasts of biology; they are contingent models produced through a combination of algorithm(s), assumptions, and available data. With this in mind, an informed consent process should not only include disclosure of biological risks but should also explicitly state all uncertainty attached to computing. Patients should be informed not only that there may be off target edits but also that there are epistemic uncertainties in the simulations that drove their therapeutic design [258].

### 9.3. The Dual-Risk Landscape: Intersections Between Biology and Computation

Co-development of CRISPR and AI collectively establishes a dual-risk landscape of impact, where a failure in either domain has the potential to detrimentally initiate the risks associated when the two domains align. For instance, an AI model may suggest edits based on computational optimum while disregarding pleotropic functions of genes and have outcomes catastrophically worse than expected. On the other hand, CRISPR can create cellular states, which may take the form of new isoforms, unexplored epigenetic states, and altered fluxes of metabolites that may not be points of knowledge built into AI model training, rendering them inadvertently blind to inducing novel risks. Relationships between synthetic biology and AI models then allow for scenarios of what can be defined as a bio-digital runaway, where the interactions from continuous algorithmic learning and experimental molecular interventions can rapidly evolve out of the extent of effective human awareness [142].

An additional underappreciated risk that can seed these cascades is algorithmic toxicity, which is not directly caused by the CRISPR tool but occurs as AI systems misinterpret biomarkers, resulting in inappropriate or poorly timed edits. For instance, noise in inflammatory markers (e.g., cytokines) following trauma can lead to misinterpretation of active tumor recurrence, prompting cryptic edits with potentially adverse effects. The emergence of this issue highlights the requirement of an unpredictable form of dual-layered oversight, independently validating biological edits and algorithmic outputs, which can stop unwanted interventions via appropriately engineered fail-safes where model outputs diverge from expected or constructed pathways. The foundation of a closed-loop CRISPR–AI therapeutic pipeline will upend regulatory classifications. Therapeutics such as pharmacological and, in certain cases, devices fall into well understood classifications defined by products that are no longer stable or iterative advances in clinical applications. As bio-digital therapeutics do evolve continuously with algorithmic learning and the changing state of the patient, risk must be unbounded by displaying, defining, generating, or supplying risk at the level of approval or relying upon engagement of an oversight model that functions in parallel with clinical use of the intervention [259].

### 9.4. Regulatory Transformation: Toward Continuous Oversight

Currently, regulatory systems that are constrained to discrete interventions, namely dynamic adaptive interventions, are inadequate using the regulatory categories realized in neurotechnologies and their biologic status with CRISPR classification understood as biologic, or gene therapy, and AI also regarded as a device. When one considers how this occurs as a closed-loop CNS context with both AI and CRISPR, a new regulatory category emerges. A potential framework could be Good Neurotechnological Practice (GNP), which would combine components from Good Manufacturing Practice (GMP) in gene therapy with the Good Machine Learning Practice (GMLP) in AI regulation. This would allow for a clear definition of rigorous auditable standards for the biological and computational components of the effect of a neurotechnological therapy [260].

Oversight will have to shift from a one-time approval to a continuous license wherein a therapy can only be sanctioned if it can be shown to be safe on an ongoing basis. This may take the form of continuous audit trails, an auditable trail that may include evidence of every model updated, every global model parameter updated, and every CRISPR designed (or versioning). Regulators would be able to evaluate the biological fidelity and computational fidelity almost in real-time, where the standards of acceptance fluctuate depending on the components of the computer at that moment in time. And given how critical the AI supports are for any CRISPR therapeutics, the oversight burden would be more than justifiable [261].

Global harmonization will be critical. Unless oversight is coordinated, there is the potential for there to be therapies provided with a wide range of oversight, wherein there is a potential that a therapy can be run uncontrolled in one jurisdiction that is a banned practice in another. This is not a situation we should leave to fate. We would even suggest a global observatory for neuroregenerative editing, very much like the pandemic type data-sharing infrastructure and systems that were developed in the wake of COVID-19. The coordinated agency could create an ongoing register of treatment outcomes, adverse events, and algorithmic performance occurring in the field, in real time, to the world. With coordinated oversight, we could prevent fragmentation from occurring but instead be a global model for other aspects of the dispensing of emerging neurotechnologies. This would offer transparency for long-term data, and with transparency comes public trust [262].

### 9.5. Equity, Access, and the Global Dimension

In addition to the technical safety considerations noted above, the introduction of CRISPR and AI also raises fairness and justice concerns. The imbalance of neurodegenerative diseases, stroke, SCI, and TBI is disproportionately larger in low- and medium-income countries, while access to the infrastructure to develop CRISPR–AI pipelines—sequencing platforms, imaging modalities, computational power, and GMP-grade manufacturing—exists almost solely in high-income countries. The paradox of equity in neuroregeneration exists, where those most affected by neurological disability are the least likely to receive therapies designed to address their neurological disabilities [263]. The digital divide worsens this inequitable status. As genomic and imaging datasets from low- and medium-income countries are still not well represented in training datasets, there is an increased probability of algorithmic error. Inequality manifests doubly: as both exclusion from receipt of therapy and, if received, as miscalibration. If we are to better this situation, it will take a focus on a tiered licensing framework, capacity–building bilateral partnerships, and a global pool of funding to be accountable for global inclusion, without which the field risks perpetuating global neurological inequity [264]. Another ethical tension is around the definition of restoration versus enhancement. The same technologies that could restore mobility after spinal injury could also, in some other situation, be used for neuro-enhancement and to pursue military or competitive goals in a coercive context. There is cultural variability here also. Normative attitudes to genetic editing of the brain vary spatially, and global governance should aim for plurality of neuroethics, not establish a single normative framework [265].

### 9.6. Data Privacy, Security, and Neuro-Identity

CRISPR–AI pipelines require genetic, transcriptomic, imaging, electrophysiological, and behavioral datasets to formulate representations of the brain; these representations are fundamentally different from primacy identifiers in medical records because they are also representations of identity. Variants from a connectomic map, neuronal firing patterns, and genetics come together to create an individual’s biometric signature of personhood; fears extend well beyond health privacy. Security breaches of such data are likely to expose vulnerabilities or predictable patterns associated with cognition and behavior [266]. In order to mitigate the risks described above, federated learning, homomorphic encryption, and secure multiparty computation should be established as standard practices in CRISPR–AI pipelines. These approaches allow AI models to learn from distributed datasets without any direct exposure of raw data; this is a balance between the analytic capacity of AI and the imperative for privacy. Privacy is not the only consideration; ownership is also critical. The digital twins being constructed using now commercialized patient data are going beyond a patient’s clinical state and are coming into commerce. To preserve a patient’s sovereignty over their digital selfhood will require regulatory acknowledgement that a computational representation of a nervous system is an expression of personal identity—and of the right of access, control, and consent afterward [267].

The ethical and regulatory issues associated with CRISPR–AI neuroregeneration should not be viewed as peripheral issues; rather, should be viewed as central to function. The risks are not only molecular; there are also risks that are computational, societal, and philosophical—off-site edits, oncogenesis and its associated pathologies, algorithmic toxicities, equity paradoxes, and digital selfhood. Tackling these tensions will require a substantial paradigm shift: from static oversight to ongoing oversight, from siloed governance to global governance, and from a narrow to a holistic front, with inclusion of biologics, computation, equity, and identity. However CRISPR–AI therapies are remembered in the long term, their regenerative efficacies and their frame of ethical and regulatory architectures facilitating this will, after all, greatly shape whatever they become [268].

## 10. Future Horizons: The Decade Ahead

The next stretch of time is going to be instrumental in altering the neuroregeneration landscape. The fields of genome editing and artificial intelligence, initially moving along parallel lines of innovation, are now converging in a single therapeutic milieu in which biological and computational processes co-evolve. The next decade will not be defined by a single revolutionary breakthrough but by the coming together of a layered, adaptable, ethically governed constellation of molecular precision, computational intelligence, bioengineering platforms, and global governance. With the emergence of this constellation, neurological repair may be able to transition from an idealistic experiment to a reproducible and standard clinical model—again, assuming that science, ethics, and equity are embedded from the start [269].

### 10.1. Entry Points: First Clinical Applications and Critical Time Windows

The most likely candidates for early clinical translations will be those conditions for which existing therapies are ineffective and the prognosis remains dismal. In this context, spinal cord injuries and GBM are still two potent candidates, although it may also be possible to consider beginning with rare neurogenetic disorders in children like Rett syndrome, Dravet syndrome, and leukodystrophies. These uniquely pressing conditions juxtapose an incredible need, a defined molecular basis, and predictable progression after diagnosis, all suitable for CRISPR targeting guided by AI. Their introduction into therapy, however, highlights a profound challenge for life course genomic stewardship—interventions that start in childhood and extend over decades of brain development will highlight the role of health systems and care providers that become the point of entry into chronic neuroregeneration paradigms [270,271].

A new area for the next decade will likely be understanding neuro-temporal windows for repair. Ontogeny, and the genius of the human experience, displays wonderful variability in chromatin accessibility, immune permissiveness, and activity in growth-associated genes as these metrics fluctuate over time after injury or disease onset during rehabilitation. AI systems that harness longitudinal multi-omics and imaging may define physiologically restricted windows when CRISPR editing could most synergistically interdigitate with plasticity and a limited degree of maladaptive scarring. Leveraging these repair critical windows may have an unprecedented effect on future pathways for regenerative interventions [272].

### 10.2. Beyond Repair: Molecular–Electrical Symbiosis

The hypothetical functional distinction that dichotomizes molecular editing from electrophysiology modulation will ultimately erode. Brain–computer interfaces, optogenetic modulators, and neuroprosthetic development will further interweave with CRISPR platforms to generate molecular–electrical hybrid systems. In this hybrid system of regeneration, AI may identify circuit instabilities following traumatic insults (e.g., hyperexcitability) or unusual synchrony amongst networks adjacent to tumors and activate CRISPR switches simultaneously to recalibrate expression profiles across targeted neuronal or glial populations. By inverse function, CRISPR-derived neurons (e.g., from glia-to-neuron conversion) can be manipulated with adaptive neuroprosthetic stimulation to produce faster locally emerging circuits from pre-existing ones. In fact, the developed world may see the emergence of hybrid regenerative therapies where genomic editing and bioelectronic feedback loops co-regulate functional recovery in the next decade [273].

### 10.3. Organoids, Living Biobanks, and Regeneration Rehearsal Platforms

Patient-derived neuronal organoids will not just be static culture objects of study; they will act and work like in vivo biobanks where CRISPR edits and AI-mediated perturbation studies can be evaluated in iterative loops. The next decade may show a fusion of organoids and digital twins culminating in concurrent revision loop platforms: co-model ecosystems that can biologically and computationally model therapy options prior to fully deploying to a patient. These systems (looking ahead) will encapsulate individual variability and allow in silico predictions to be stress-tested against in vitro truths [274].

Integration with organ-on-chip technology may allow prediction and/or simultaneous impact through the simultaneous modeling of vascular, immune, and metabolic conditions. In this manner, regenerative interventions can be thought of as whole-system phenomena as opposed to neuron-centric occurrences. These platforms will catalyze discovery, improve personalization, and create an essential link from preclinical development into clinical trial readiness [275].

### 10.4. Adaptive Clinical Trials and Living Protocols

Traditional trials were not designed for therapies that provide ongoing innovation. We may observe the emergence of living protocols over the next decade, wherein the design of the trial is ceaselessly amended throughout the course of the trial in response to developing data. Patients may not receive specific interventions; rather, patients may receive adaptive n-of-1 journeys where CRISPR strategies and real-time AI navigation are iteratively optimized in response to other functional outcomes and/or the dynamics of biomarkers [128]. These frameworks will require new forms of ethics and regulation. Informed consent must consider that the therapeutic trajectory may not only be idiosyncratic but also evolve rapidly, with algorithms recalibrating interventions over time. However, the potential is revolutionary—rather than having static, population-derived endpoints, neuroregeneration trials may become personalized, adaptive, and longitudinally optimized—a true reversal of the clinical trial paradigm [276].

### 10.5. Expansion to Neuro-Immune and Neuro-Metabolic Axes

Neuroregeneration will emerge as we expand beyond the neuron-centric repair in favor of systemic axes, which may contribute profoundly to neuroregenerative outcomes. The neuro-immune interface is increasingly acknowledged to be both a barrier and a linchpin of repair with microglial states, T-cell infiltration, and peripheral immune arm crosstalk directing regenerative impact. CRISPR may be utilized to reprogram maladaptive microglial phenotypes or to modulate chronic neuroinflammation, while AI tracks immune signatures in order to signal flare states or maladaptive cascades [277]. In this fashion, neuro-metabolic resilience may emerge more urgently. Mitochondrial dysfunction, nutrient sensing dysregulation, and exhaustion as determinants of capacity for repair. AI models could also combine metabolic flux analysis with neural imaging to create targeted CRISPR edits that would provide systemic metabolic support for regeneration. This transition from a narrower focus on neuron circuits to a much broader focus on the whole organism indicates a sensible progression whereby new semantics surrounding neuroregeneration can emerge in the next decade [278].

### 10.6. Translational Readiness and Scalable Manufacturing

Shifting from confirmed laboratory viability to confirmed translational efficacy for clinical use is a multi-factorial process; subsequent bottlenecks for CRISPR–AI therapies will require proposals for the manufacture, scaling, and automation. The CRISPR–AI therapeutic paradigm will necessitate the usage of GMP-grade production lines for viral vectors, nanoparticles, or exosome-based carriers, which also require strict safety and quality assurance checks with good manufacturing process specifications [279]. AI-generated automation could also increase the design and testing of sgRNAs, aid in optimized packaging of the vector, and be able to design stochastic testing to predict potential off-target landscapes, which will decrease timelines for development from months to days.

Distributed manufacturing hubs may develop whereby the therapies are able to develop and not be limited by the few academic centers and be produced in a sustainable fashion through regionally produced manufacturing facilities. This distribution will support trials initiated through population diversity and alleviate some of the inequities that accompany a centralized manufacturing approach. The trajectory in a decade will represent a major shift away from bespoke lab interventions and toward interventions that are scalable, reproducible, and manufacturable with globalized therapeutic applications [280].

One of the least appreciated challenges of CRISPR–AI neuroregeneration is not conceptual advancement but a flow of resources across biological, computational, and clinical medicine. The pipeline has structural differences from what would be considered normal therapeutic mechanisms in that they rely on nebulous collaborations between genome-editing laboratories, high performance computational frameworks, and clinical trial facilities. Each niche network has unique cultures, operating standards, and incentive systems; molecular nodes that focus only on target validation and vector creation; AI nodes that need massive curated datasets; and clinical nodes that are concerned about patient safety, regulatory, and outcomes issues [281]. To the point they furthermore do not share the same goals. One clinical hospital has databases and longitudinal studies but does not have encrypted computational pipelines for multiparameteromics and imaging; one AI research lab has statistical procedures for analysis of outcomes—the validation was not performed in time. Although, some versions of procedures performed on each node are well resourced, the international regulatory systems, incompatible formats, and fragmented or proprietary ownership of IP, especially if trialed in early stages of translation, all conspire to stretch timelines of translation [282].

These issues would best be addressed with a concerted structure of interdisciplinary fusion. This could involve federated sharing networks to protect patient privacy while being able to train AI models in geographic cohorts; consortia of translational teams who embed computational scientists at or into molecular and clinical teams; and trials that have modular operations that work in a coordinated manner to shuffle laboratory developments and algorithm updates through ethically governed structures [283]. Beyond logistics, it is a culture problem: interdisciplinary training cultures where professionals develop as translators of genome editing and computational modeling to migrate in and outside of fields or disciplines. Establishing investments in these types of constructs would be pivotal to avoid CRISPR–AI therapeutics as pilot projects. It would be much more desirable to find CRISPR–AI therapeutics within clinical protocols that are scalable, reproducible, and equitable frameworks that are able to collaboratively and seamlessly maintain technical integrity and coordination [284].

### 10.7. Planetary Dimensions and Sustainability

The intersection of CRISPR and AI is not without regard for planetary context. The burden of neurological disease will be continually influenced by wide-ranging global demographic shifts, climate-induced changes in the incidence of stroke and trauma, and health infrastructures that are variable in regard to availability and access. The planetary ethics of neuroregeneration will require a delicate balancing act: detecting leading edge innovation while managing sustainable practices, i.e., reducing the carbon footprint of AI training, securing the raw materials to manufacture vectors, and ensuring equitable access across continents [285].

Within a decade, we may galvanize a global neuroregeneration observatory that would organize data sharing, safety reporting, and ethical oversight across regions. Models of open source AI that have been trained on globally diverse databases, and that can provide distributed GMP hubs for vector production may address the significant disparity between the access to these technologies in regions across the globe. Otherwise, we run the risk of the technological advances actually amplifying neurological inequity rather than solving it [286].

### 10.8. Long-Term Stewardship and Neuro-Identity

Perhaps the most challenging aspect of raising within the next decade will not be the delivery of the therapies but rather, the stewardship of such therapies. For example, CRISPR edits in the nervous system may last a long time (i.e., decades), which would also decay and lose efficacy in conjunction with aging, comorbidities, and environmental exposures, some of which could be unknowable. AI could help establish a digital neuro-stewardship framework that would provide lifelong longitudinal monitoring and assess data streams of electrophysiology, imaging, cognitive assessments (e.g., integrated psychometric experiments), and omics data to perhaps assess or ‘track’ neural repair trajectories long after the intervention [259]. Moreover, those digital twins can be thought of, once built from neural data, as not only digital representations of self but also bio-computational representations, and questions of neuro-identity and neuro-sovereignty will need to consider an increasingly virtual existence. Patients may need rights not only to their biological edits but also to any computational representation of their selfhood. A neuro-CRISPR registry could also be established, like a similar registry for oncology, to help provide these patients with transparency, continuous oversight, and collective knowledge of others’ clinical experiences [287]. The next decade will not be measured by milestones, but rather the concomitant threading together of trajectories: recognizing repair-critical time windows, developing molecular–electrical systems in symbiosis, establishing organoid rehearsal platforms, utilizing living trial protocols, and expanding the object of immune and metabolic periods, compounded by the issues of scaling manufacturing for dimensionality and planetary sustainability, and long-term digital stewardship [288].

All these strands, together, provide a vision where neuroregeneration is grounded in living ethical sourcing, technical perfection, and planetary equity or fortitude. The ultimate success of this trajectory will not only rest on whether or not we can restore loss of function but rather how successfully we can leverage fairness, equity, transparency, and sustainability as the central tenets of innovation itself. If we can do this, we might be able to think of the next decade as approximately the time we transacted neuroregeneration from a wished outcome of remaining experimental into a durable clinical and ethical platform.

## 11. Conclusions

Restoration of the CNS has historically ranked among the most intractable medical dilemmas based on biological constraints (e.g., innate inhibitors of growth), hostile microenvironments disrupted by tumor or injury, and the intricate complexity of neural circuits. Recent advances in genome editing and computing promise to perhaps catapult those constraints into the ‘likely’ category. CRISPR technologies presently allow targeted, specificity-mediated disruption of inhibitory pathways, epigenetic landscapes, and tumor or injury microenvironments, while artificial intelligence enables the fusion of multiple biological and clinical datasets into adaptable and predictive frameworks. Taken together, these advances do not eliminate the difficulty of neuroregeneration, but they do begin to describe how future therapies might move away from fixed interventions toward adaptable and individualized processes. This synthesis describes CRISPR’s ability to reactivate previously dormant regenerative programs, shift cellular lineages, and recalibrate immune and metabolic responses, while AI offers the ability to further dissect such heterogeneity, model intervention outcomes, and adapt therapies. The convergence of these sciences signals an important emerging paradigm of precision neuroregeneration in which the design of therapy is iterative, biologically anchored, and computing determined projection of patient-specific trajectories.

Nevertheless, challenges remain significant. The field of genome editing still raises questions around mosaicism, stability, and long-term safety. Artificial intelligence is now encountering questions of comingling sources of bias, risk of algorithmic opacity, and inequity in reproducibility across different bioethics foundations. Access equity, the sustainability of therapeutic production, and stewardship of digital representations of patients will equally influence the direction of the field as much as scientific innovation. The decade ahead will require we frame scientific discovery, clinical practice, ethical contemplation, and policy design together as co-determinative rather than independent pathways. Innovation is likely to remain incremental, by way of many surprising and repeated discoveries, expanding the concept of possibility rather than suddenly generating transformation. Early adopters may appear in glial repair, GBM, or rare neurogenetic syndromes, but the stronger body of work will support an increasingly adaptive regenerative medicine, using systemic axes of immune, metabolic, cognitive, and patient biology time-dependent factors as principles to synchronize intervention planning. The takeaway from this review is not meant to be a final word but an attempt to map out a quickly changing landscape and point out where the intersection of molecular and computing sciences may be most productive. While neuroregeneration is still an unfinished project, I hope that the considerations described here continue the conversation, both encouraging experimentation and critique within, as well as collaboration across disciplines. If these perspectives lead to any development of safer genome editing platforms, more transparent algorithms/treatments, or fairer models of access, I am content.

Thus, the conclusion to this paper is not an end but a beginning. The decade ahead will reveal if the forces of CRISPR and artificial intelligence become mature clinical and ethical paradigms that decentralize medicine. For now, with caution but a positive outlook, questions that were once considered to lie beyond mere neuroscience are starting to take shape as an experimental, tangible form. The charge will be to proceed with humility and care as to when and how it leads us forward.

## Figures and Tables

**Figure 1 ijms-26-09409-f001:**
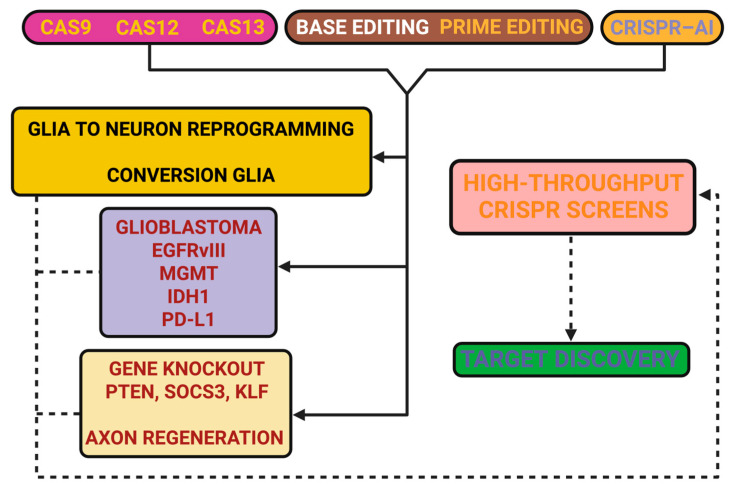
CRISPR toolbox in neuroregeneration. Figure 1 aims to outline representative CRISPR modalities and their potential applications in CNS repair. Cas9 nucleases have been applied for knockout of intrinsic inhibitors such as PTEN and SOCS3, thereby facilitating axonal growth and plasticity. Cas12a nucleases provide broader PAM recognition and have been explored for editing extracellular matrix regulators, such as sulfotransferases, to reduce chondroitin sulfate proteoglycan–mediated inhibition. Cas13 platforms enable transient RNA targeting, which has been used to silence pro-inflammatory transcripts (e.g., NLRP3, IL-1β) and to downregulate MGMT in GBM. Base editors (CBE, ABE) have been tested for precise correction of mutations, including SOD1 in amyotrophic lateral sclerosis and tau-related modifications in Alzheimer’s disease. Prime editors extend editing to small insertions and deletions, with examples in MECP2 (Rett syndrome) and SCN1A (Dravet syndrome). CRISPRa/i strategies allow transcriptional modulation without cleavage, with applications such as glia-to-neuron conversion (Ascl1, NeuroD1 activation) and repression of A1 astrocytic genes (C3, Serping1). The figure also incorporates GBM as a context in which targets such as EGFRvIII, IDH1, PD-L1, and MGMT are being investigated. High-throughput CRISPR screens in organoids and stem cells are included as a discovery pipeline feeding into these applications. The schematic is intended as a concise visual complement to the text, highlighting representative approaches rather than offering an exhaustive catalog.

**Figure 2 ijms-26-09409-f002:**
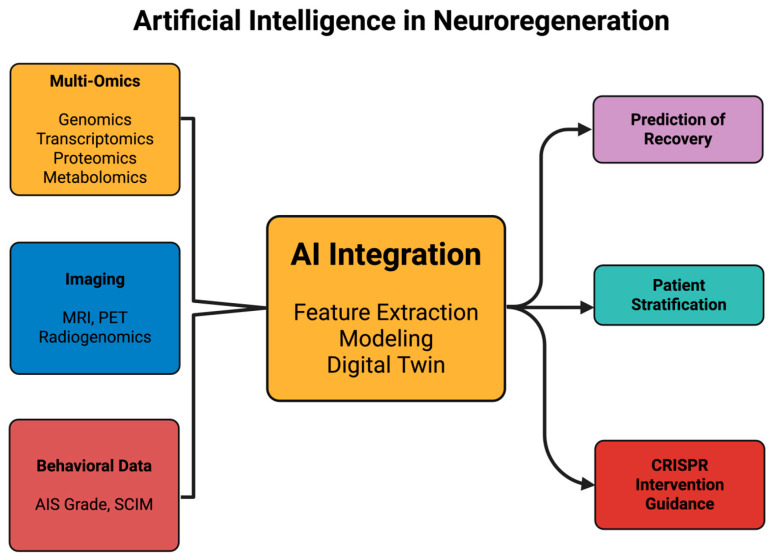
Artificial intelligence in neuroregeneration. Figure 2 aims to illustrate how artificial intelligence functions as an integrator of diverse biological and clinical datasets in the context of neuroregeneration. Inputs include single-cell and spatial multi-omics (such as scRNA-seq and ATAC-seq), imaging modalities (MRI, PET, and radiogenomics), electrophysiological recordings, and standardized recovery assessments (e.g., AIS grade, SCIM). These data streams converge into AI-driven analytical frameworks, including deep learning, graph neural networks, and patient-specific digital twin models. The outputs encompass predictive recovery models after spinal cord injury, radiogenomic classification in GBM, therapeutic target prioritization for genome editing strategies, and adaptive clinical trial design. A closed-loop feedback layer highlights how patient outcomes can be continuously integrated back into AI models, refining predictions and supporting real-time therapeutic adjustments. The schematic is intended as a conceptual overview, emphasizing representative applications rather than providing an exhaustive catalog.

**Table 1 ijms-26-09409-t001:** Principal molecular and cellular barriers to CNS repair, with recent evidence from 2024 to 2025 studies and potential strategies for modulation. The table summarizes representative mechanisms and translational levers, offering a concise overview rather than a comprehensive catalog.

Barrier/Node	Cellular Context	Mechanistic Constraint	Key Findings (2024–2025)	Actionable Levers	Biomarkers and Readouts	Evidence Level	References
Nogo-A/NgR1 Axis	Oligodendrocyte myelin → neurons	Myelin-associated inhibition of axonal growth; growth cone collapse	Phase 2b RCT of anti-Nogo-A (NG101) in cervical SCI: motor-incomplete patients showed improved upper-limb function; no effect in complete SCI	CRISPR editing of RTN4 or NgR1; biologics (anti-Nogo-A); rehab-timed dosing	UEMS, SCIM, AIS grades; plasma Nogo-A; diffusion MRI; AI responder profiling	Clinical trial (Phase 2b)	[57]
CSPGs/Perineuronal Nets	Reactive astrocytes; ECM around PV interneurons and CA2 pyramids	Steric blockade and receptor-mediated inhibition (PTPσ/LAR)	PTPσ blockade restores autophagy flux and synaptic markers; PNNs constrain plasticity in CA2/PV circuits; microglia reduce PNN density	CRISPR targeting of PTPRS or sulfotransferases (CHSTs); ChABC enzymes; biomaterials	CS-GAG sulfation profiles; GAP-43 staining; fMRI plasticity; AI mapping of PNN density	Preclinical (rodent, in vivo)	[58,59,60]
DREZ Exclusion	Peripheral–CNS interface	Sensory axons fail to enter spinal cord; intrinsic growth ceiling	PTEN deletion + kaBRAF activation enables axons to cross DREZ; SOCS3 deletion alone ineffective	CRISPR combinatorial editing (PTEN↓ + BRAF tuning); neuromodulation	Axon counts past DREZ; pS6/ERK; proprioceptive evoked potentials; AI tracing	Preclinical (rodent)	[61]
Neuron-Intrinsic Growth Brakes (PTEN, SOCS3, KLFs)	Mature CNS neurons	mTOR/STAT repression; gene silencing	Syt4 inhibition blocks corticospinal sprouting; PTEN+BRAF synergy stronger than SOCS3 deletion	CRISPRi for Syt4; base/prime editing of KLFs; conditional PTEN modulation	CST axon density; ladder rung/gridwalk; calcium imaging; AI kinematic prediction	Preclinical (rodent)	[62,63]
Reactive Astrocytes/Glial Scar	Astrocytes, perivascular fibroblasts	A1 neurotoxic signaling; ECM deposition	Microglial exosomes suppress A1 astrocytes (murine SCI); small molecules promote A1→A2 shift; NSC-derived astrocytes create pro-regenerative niches	CRISPR editing of C3/Serping1; dCas9-KRAB/VP64 for astrocyte polarization; engineered exosomes	GFAP/vimentin borders; C3/C1q markers; scar topology (3D imaging); AI scar segmentation	Preclinical (rodent, in vitro)	[64,65,66]
Microglial Activation and Pyroptosis (NLRP3 Axis)	Microglia, infiltrating myeloid cells	Inflammasome-driven pyroptosis; maladaptive pruning	Microglia depletion reduces aberrant sprouting and autonomic reflexes; NLRP3 targeting improves outcomes (2025 reports)	CRISPR disruption of NLRP3, CASP1, RelA; immune-targeted delivery; small-molecule inhibitors	Autonomic reflex testing; IL-1β/IL-18; scRNA-seq microglia states; AI classifiers	Preclinical (rodent)	[67,68]
PNN-Linked Circuit Closure	PV interneurons, hippocampal CA2	Restriction of synaptic remodeling	CA2/PV PNNs constrain plasticity; microglia dismantle PNNs and reopen plasticity	CRISPR targeting CHST3/CHSY1; PTPRS modulation; timed remodeling with rehab	WFA-labeled PNNs; PV tuning curves; EEG/MEG plasticity indices; AI image analysis	Preclinical (rodent)	[58,59]
Integrated Stress Response (ISR/eIF2α)	Neurons, glia	Translational arrest; apoptosis; slow repair	ISRIB attenuates eIF2α-P, reduces neuronal death, improves recovery (rodent SCI)	CRISPR modulation of PERK/GCN2; transient CRISPRi; ISRIB adjunct	p-eIF2α, CHOP/ATF4; NeuN survival; AI recovery scores	Preclinical (rodent)	[69]

**Table 2 ijms-26-09409-t002:** CRISPR modalities with emerging applications in neuroregeneration. The table summarizes major platforms, their underlying principles, and selected recent studies. Included examples illustrate potential therapeutic levers and measurable outputs, providing a concise reference point for ongoing experimental and translational work.

CRISPR Platform/Variant	Mechanistic Principle	Emerging Applications	Representative Evidence (Model → Key Finding)	Source
SpCas9/SaCas9 nuclease	Double-strand breaks for gene knockout	PTEN (axonal growth), NgR1 (myelin inhibition), EGFRvIII/PD-L1 (GBM)	Rodent SCI: PTEN knockout promotes axon regrowth; GBM organoid: Cas9-mediated EGFRvIII + PD-L1 KO reduces invasiveness	[57,100,101]
Cas12a (Cpf1)	Staggered DNA breaks, flexible PAM targeting	CSPG sulfation enzymes (ECM remodeling); HIF1α (tumor angiogenesis)	Zebrafish SCI: CSPG sulfotransferase KO via Cas12a restores axon extension; Mouse GBM: HIF1α enhancer editing reduces angiogenesis	[60,102]
Cas13 (RNA-targeting)	Programmable RNA knockdown/editing	NLRP3 (inflammation), MGMT (chemoresistance)	CRISPRi via Cas13 diminishes inflammasome activation in SCI; targeting MGMT mRNA resensitizes GBM cells	[103]
Base Editors (CBE/ABE)	Precise base conversions without DSBs	SOD1 (ALS), tau splicing (AD), HTT expansions (Huntington’s)	ABE correction of SOD1 mutation preserves motor neurons in mouse ALS model	[104]
Prime Editors	Search-and-replace precision editing	MECP2 (Rett), SCN1A (Dravet), HTT expansions	Prime editing corrects MECP2 in Rett iPSCs, rescues gene expression and synaptic markers	[105]
CRISPRa/CRISPRi (dCas9)	Transcriptional modulation (activator/repressor)	STAT3 (axon growth), Ascl1 (glia-to-neuron), astrocyte polarization	CRISPRa-STAT3 accelerates regeneration in mouse SCI model; dCas9-VPR Ascl1 converts astrocytes to neurons	[106]
High-throughput CRISPR screens	Functional genomics for target discovery	Regeneration regulators; GBM adaptive resistance	Organoid CRISPR-seq pinpoints cortical neuron sprouting genes; GBM screens find resistance mechanisms	[97]

**Table 3 ijms-26-09409-t003:** Applications of artificial intelligence in neuroregeneration (2024–2025). The table outlines major domains, methodological models, experimental and clinical contexts, measurable outputs, and translational implications, together with representative references.

AI Domain	Method/Model	Function in Neuroregeneration	Specimen Model and Use Case	Key Outputs/Biomarkers	Translational Relevance	References
Single-cell/spatial omics integration	Graph neural networks + trajectory inference	Identify regenerative subpopulations; define inhibitory microenvironments	Mouse SCI atlas integrating scRNA-seq + spatial transcriptomics	Vsx2+ interneuron emergence; reactive astrocyte states	Identifies repair-competent subtypes; potential CRISPR targets for reprogramming	[132]
Predictive recovery modeling in SCI	Ensemble ML; random forest; survival analysis	Forecast patient-specific recovery trajectories	Clinical SCI datasets (N > 1500)	AIS grade transition; motor score prediction	Guides individualized rehab and stratifies patients for trials	[133]
Radiogenomics in GBM	CNN + radiogenomic classifiers	Infer oncogenic pathway activity from imaging	MRI datasets linked to molecular profiling	RTK-RAS, PI3K, TP53 activity inferred	Enables non-invasive biopsy surrogates; aids AI-CRISPR targeting	[134]
Deep learning radiomics	3D CNN + radiomic feature extraction	Predict progression and therapy response	Multiparametric MRI in GBM	Tumor growth velocity, invasion index	Improves adaptive treatment planning	[135]
Multimodal survival prognostication	Integrative AI (clinical + imaging + omics)	Predict OS/PFS in glioma patients	Glioma cohorts (n = 400)	Hazard ratios, individualized survival curves	Precision trial design; long-term prognosis	[136]
Histopathology image AI	CNN trained on TCGA slides	Grade and molecular marker prediction	Glioma H&E datasets	IDH mutation status; MGMT methylation probability	Digital pathology decision support	[137]
Nanotheranostic synergy	AI + nanocarrier design; CRISPR payload optimization	Enhance BBB penetration and targeted delivery	GBM nanocarrier studies	Nanoparticle biodistribution; editing efficiency	Improves CRISPR delivery precision	[138]
Therapy response prediction	Radiomics + ML classifiers	Identify responders early in treatment	GBM/BM imaging datasets	Delta-radiomics signatures	Enables rapid therapeutic switching	[139]
Research horizon mapping	Scientometric + bibliometric AI	Identify hotspots in AI-SCI research	Global publication datasets (2014–2024)	Keyword co-occurrence, clustering	Anticipates future research shifts (BCI, robotics)	[140]

## Data Availability

The data presented in this study are available on request from the corresponding author.

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
