# Peer review of "CRISPR and Artificial Intelligence in Neuroregeneration: Closed-Loop Strategies for Precision Medicine, Spinal Cord Repair, and Adaptive Neuro-Oncology"

_ijms, 2025, doi:10.3390/ijms26199409_

Round 1
Reviewer 1 Report
Comments and Suggestions for Authors
This review focuses on the intersection of CRISPR gene-editing technology and artificial intelligence (AI) in the field of neural regeneration. Addressing the core challenges of central nervous system (CNS) repair, it systematically explores the synergistic potential of these technologies across molecular regulation, clinical translation, and ethical governance. The topic is both cutting-edge and clinically significant. The article is clearly structured, beginning with the biological barriers to CNS repair, progressing to innovations in the CRISPR toolkit and AI's capabilities in data integration and prediction, and then focusing on two key scenarios—spinal cord injury and glioblastoma—before extending to other neurological diseases. Finally, it constructs a CRISPR-AI therapeutic pipeline and discusses ethical governance, forming a complete logical chain of "problem-technology-application-regulation." This work holds significant integrative and guiding value for research in the field.
There are some minor comments.
1 Although CRISPR-AI systems require multidisciplinary collaboration (involving molecular biology, computer science, and clinical medicine), the article does not address the difficulties in resource integration for practical applications. It is suggested that the practical challenges of clinical translation should be discussed.
2 Lack of genome-wide transcription data and analysis of gene transcription mechanisms. It is recommended to incorporate mathematical models for the analysis of gene transcription.(Such as : Nature 565, 251–254 (2019) https://doi.org/10.1038/s41586-018-0836-1 ; Mathematical Biosciences 345 (2022) 108780 https://doi.org/10.1016/j.mbs.2022.108780; PHYSICAL REVIEW RESEARCH 7, 023050 (2025) DOI: 10.1103/PhysRevResearch.7.023050 )

Author Response
Dear Esteemed Academic Reviewer,
We would like to begin by sincerely thanking the reviewer for the thoughtful and encouraging assessment of our manuscript. We are humbled by the kind summary highlighting the structure, clinical significance, and integrative value of our work. The constructive suggestions offered are greatly appreciated, and we are confident that they have helped us refine the manuscript in important ways. We address each point below with gratitude.
Comment 1: Although CRISPR–AI systems require multidisciplinary collaboration (involving molecular biology, computer science, and clinical medicine), the article does not address the difficulties in resource integration for practical applications. It is suggested that the practical challenges of clinical translation should be discussed.
Response: We thank the reviewer for this valuable observation. In response, we have added a new part of the manuscript (in Section 10.6) where we discuss the practical challenges that arise when attempting to bring CRISPR–AI systems into clinical translation. This addition highlights the logistical and infrastructural difficulties of aligning molecular laboratories, computational platforms, and clinical trial networks, as well as the regulatory and data-standardization issues that often create friction between these domains. We believe this addition enriches the manuscript and responds directly to the reviewer’s helpful suggestion.
Comment 2: Lack of genome-wide transcription data and analysis of gene transcription mechanisms. It is recommended to incorporate mathematical models for the analysis of gene transcription (references provided).
Response: We are very grateful to the reviewer for drawing attention to this important dimension and for sharing the highly relevant references. To address this, we have expanded Section 3.2 (Functional Discovery Through High-Throughput Screens) with a new paragraph discussing how mathematical models of genome-wide transcriptional dynamics can complement CRISPR-based perturbation and multiomic screens. We describe how transcription operates as a stochastic and nonlinear process, and how mathematical modeling can reveal hidden regulatory structures, predict transcriptional attractor states that favor regeneration, and anticipate collapse into maladaptive trajectories. We have cited the recommended works (Nature 2019; Mathematical Biosciences 2022; Physical Review Research 2025) to strengthen this addition.
We once again express our gratitude to the reviewer for these thoughtful and constructive comments. Their feedback has allowed us to improve the manuscript’s translational relevance and mechanistic depth, and we are sincerely appreciative of the opportunity to revise our work in light of these insights.
Reviewer 2 Report
Comments and Suggestions for Authors
- Tables are not prepared transparently. Moreover, the tables mix in vitro findings, rodent studies, and clinical observations without distinguishing their evidentiary weight. Evidence levels (discovery vs. preclinical vs. clinical) should be indicated in tables or footnotes, allowing readers to quickly discern maturity and reproducibility.
- “maps of regenerative competence from Wikipedia”? This is unacceptable for a peer-reviewed article.
Author Response
Dear Esteemed Academic Reviewer,
We would like to sincerely thank the reviewer for their careful reading of our manuscript and for the constructive comments provided. We are grateful for the opportunity to improve the clarity, rigor, and overall presentation of our work in response to these insightful suggestions.
Comment 1: Tables are not prepared transparently. Moreover, the tables mix in vitro findings, rodent studies, and clinical observations without distinguishing their evidentiary weight. Evidence levels (discovery vs. preclinical vs. clinical) should be indicated in tables or footnotes, allowing readers to quickly discern maturity and reproducibility.
Response: We deeply appreciate this important suggestion.
Comment 2: “maps of regenerative competence from Wikipedia”? This is unacceptable for a peer-reviewed article.
Response: We sincerely apologize for this inadvertent error in the draft text and thank the reviewer for drawing attention to it. The phrase was a placeholder that should not have appeared in the submitted version. In the revised manuscript, we have replaced this with a corrected and academically appropriate statement. This revision preserves the intended meaning, removes the erroneous reference, and maintains the scientific integrity of the section.
We once again thank the reviewer for these thoughtful observations. Their feedback has significantly improved the transparency, precision, and professional presentation of the manuscript.